# DRAW-AND-UNDERSTAND: LEVERAGING VISUAL PROMPTS TO ENABLE MLLMS TO COMPREHEND WHAT YOU WANT

**Weifeng Lin[1]*  Xinyu Wei[2]*  Ruichuan An[2]  Peng Gao[4]  Bocheng Zou[3]**

**Yulin Luo[2]  Siyuan Huang[4]  Shanghang Zhang[2]  Hongsheng Li[1]†**

[1]CUHK  [2]Peking University  [3]University of Wisconsin–Madison  [4]Shanghai AI Laboratory

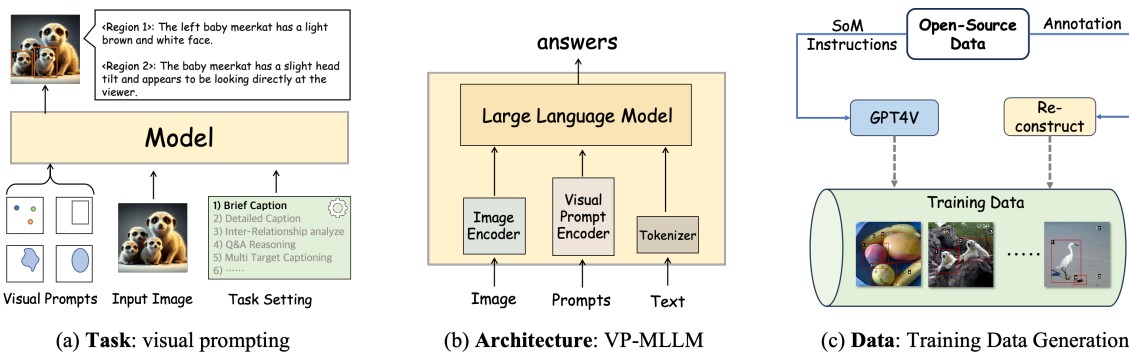

(a) **Task**: visual prompting  (b) **Architecture**: VP-MLLM  (c) **Data**: Training Data Generation

Figure 1: **Overview of the Draw-and-Understand Framework. (a)** Illustrating the task of visual prompting understanding. **(b)** The architecture of Visual Prompting MLLM (VP-MLLM), which consists of an image encoder, a visual prompt encoder, and an LLM. **(c)** The data generation process for training, which involves two components: reconstruction of open-source data and data generation assisted by GPT-4V.

## ABSTRACT

In this paper, we present the **Draw-and-Understand** framework, exploring how to integrate visual prompting understanding capabilities into Multimodal Large Language Models (MLLMs). Visual prompts allow users to interact through multi-modal instructions, enhancing the models' interactivity and fine-grained image comprehension. In this framework, we propose a general architecture adaptable to different pre-trained MLLMs, enabling it to recognize various types of visual prompts (such as points, bounding boxes, and free-form shapes) alongside language understanding. Additionally, we introduce MDVP-Instruct-Data, a multi-domain dataset featuring 1.2 million image-visual prompt-text triplets, including natural images, document images, scene text images, mobile/web screenshots, and remote sensing images. Building on this dataset, we introduce MDVP-Bench, a challenging benchmark designed to evaluate a model's ability to understand visual prompting instructions. The experimental results demonstrate that our framework can be easily and effectively applied to various MLLMs, such as SPHINX-X and LLaVA. After training with MDVP-Instruct-Data and image-level instruction datasets, our models exhibit impressive multimodal interaction capabilities and pixel-level understanding,

---

*Equal Contribution
†Corresponding Authors

while maintaining their image-level visual perception performance. The code and related resources are available at `https://draw-and-understand.github.io`.

# 1 INTRODUCTION

Recent works (Liu et al., 2024b; Zhu et al., 2023; Liu et al., 2023a; Bai et al., 2023; Liu et al., 2024a) have enhanced Large Language Models (LLMs) with visual perception, facilitating image-related communication and fostering a deeper understanding of the world. These models primarily focus on interpreting whole images by aligning them with text prompts. However, simple language interactions often fail to capture users' true intentions, especially when users need to highlight specific areas in images that are difficult to describe with words. Consequently, there is growing interest in enabling visual prompting capabilities in Multimodal Large Language Models (MLLMs) to enhance interactivity and pixel-level understanding.

To achieve this, ChatSpot (Zhao et al., 2023) and Shikra (Chen et al., 2023b) first utilize textual representations to specify coordinates within images, thereby enhancing interaction with the models. Some studies (Peng et al., 2023; Zhang et al., 2023a; Zhou et al., 2023) employ positional embeddings to improve spatial recognition, while others (Rasheed et al., 2023; Zhang et al., 2023a; You et al., 2023; Yuan et al., 2024a; Chen et al., 2023a) focus on extracting Regions of Interest (ROI) to enhance attention to specific areas of images. Additionally, LLaVA-ViP (Cai et al., 2023) introduces visual markers to facilitate more intuitive interactions between users and models. (More related works are discussed in Sec. B)

However, existing methods have several limitations: $(i)$ ROI-based methods (Zhang et al., 2023a; Rasheed et al., 2023; Yuan et al., 2024a; You et al., 2023) are typically designed for specific architectures. For example, they often rely on pre-attached segmentation models or externally provided ground truth masks, which hinders user flexibility and model scalability. Additionally, they require training from scratch, leading to substantial resource consumption; $(ii)$ Most methods (Zhang et al., 2023a; Rasheed et al., 2023; Zhao et al., 2023; Chen et al., 2023b; Peng et al., 2023) depend on fixed visual references, such as bounding boxes, which are neither flexible nor user-friendly; $(iii)$ Several methods (Yuan et al., 2024a) fail to support the simultaneous referencing of multiple objects, limiting their flexibility and preventing them from addressing more complex understandings, such as nuanced interrelations and spatial dynamics with surrounding entities and backgrounds; $(iv)$ Most methods (Zhang et al., 2023a; Rasheed et al., 2023; Zhao et al., 2023; Chen et al., 2023b; Peng et al., 2023; Yuan et al., 2024a; You et al., 2023) primarily focus on visual prompting understanding but neglect image-level perception performance, which limits their practical applicability.

To address these challenges, we present the Draw-and-Understand framework, a solution specifically designed to endow multi-modal LLMs with visual prompting understanding. As depicted in Fig.1(a), we first define the form of the visual prompting understanding task, emphasizing the multi-modal interaction between users and the model. Next, we introduce visual prompting MLLM (VP-MLLM), a general architecture that comprises a vision encoder, a visual prompt encoder, and an LLM. This architecture can be efficiently adapted to most mainstream pre-trained MLLMs, enabling them to easily acquire visual prompting capabilities while maintaining their original robust vision-language understanding. Notably, our proposed visual prompt encoder enhances the model's referring capability by embedding the coordinate features of visual prompts. It supports multiple input formats and can process several visual prompts simultaneously.

In addition, instruction tuning data for visual prompting is an essential component. To this end, we curate a comprehensive Multi-Domain Visual Prompt Instruction Dataset (MDVP-Instruct-Data). This dataset comprises image-point-text and image-region-text pairs, totaling approximately 0.8 million images and 1.2 million query-answer triplets. We compiled this dataset by integrating existing datasets containing segmentation masks or bounding box annotations, while leveraging GPT-4V's advanced image understanding capabilities for data generation. MDVP-Instruct-Data provides detailed attribute information for objects identified by visual prompts, including their relationships with surrounding entities and the background. This dataset

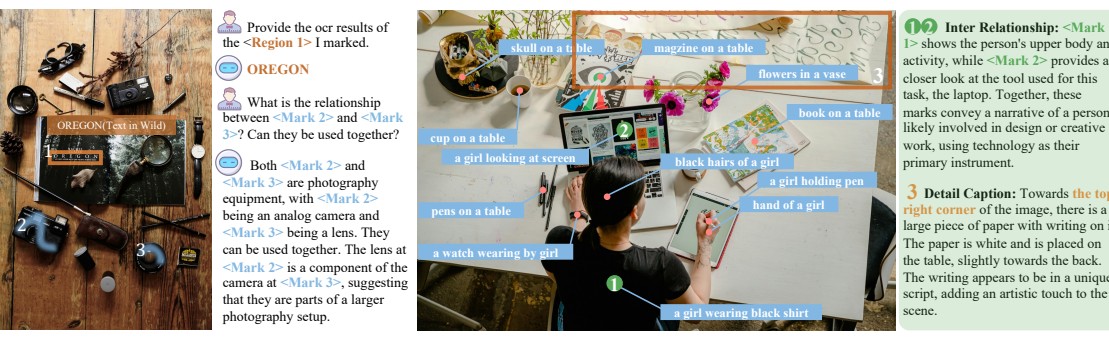

Figure 2: **(Left)** Our approach achieves unprecedented performance in OCR tasks and excels at understanding implicit relationships and conducting complex reasoning. **(Right)** Our approach demonstrates the capability to perceive objects at a pixel level granularity, such as 'a girl holding a pen'. Furthermore, it can perform detailed captioning and inter-relationship analysis on arbitrarily shaped Regions Of Interest (ROIs).

enhances spatial understanding across various image domains, thereby improving the robustness of model's responses in diverse visual contexts. With these approaches in place, users can interact with the model using their native language and refer to specific actions, such as clicking and drawing, to obtain desired answers about the region of interest. This allows the MLLMs to engage with the physical world more effectively.

To assess the strength and resilience of visual prompting models, we introduce MDVP-Bench, a benchmark designed to evaluate visual prompting comprehension abilities. MDVP-Bench encompasses a variety of tasks, including point-level and region-level captioning, inter-relationship analysis, and complex reasoning. In evaluating our VP-MLLMs alongside other visual prompting methods on MDVP-Bench, we find that ours consistently outperforms the other methods. We anticipate that MDVP-Bench will provide a solid foundation for future research in the field of visual prompting and multimodal models.

## 2 MDVP-INSTRUCT-DATA

In this section, we introduce the Multi-Domain Visual Prompt Instruction Dataset (MDVP-Instruct-Data), designed to enhance interactivity and fine-grained image understanding in MLLMs. It primarily consists of two types of data:

**(1) Dynamic Multi-Target Captioning.** In this context, users are assumed to provide $N$ visual prompts to refer to regions of interest, and the model generates a comprehensive caption describing all $N$ subjects. To create this type of data, we observed that datasets related to Referring Expression Comprehension (REC) and Phrase Grounding can be transformed to align with this task. Specifically, these datasets include the ability to localize visual content and ground each entity mentioned by a noun phrase in the caption to a specific region in the image. Building on this insight, we inverted the inputs (captions) and outputs (coordinates of subjects), treating the ground truth boxes as input visual prompts, with the corresponding natural language descriptions serving as the answers. The public datasets we utilized include Flickr30K (Plummer et al., 2015), RefCOCO/+(Yu et al., 2016), GCG(Rasheed et al., 2023), and GRIT (You et al., 2023), as well as GeoChat (Kuckreja et al., 2023) from remote sense domain.

**(2) Instruct-Conversation Data.** For visual prompting conversations, we primarily employ two methods for data construction: *(i) Reconstructing from Grounding QA datasets*. Grounding QA datasets provides ground truth bounding boxes for various target objects in both the questions and answers. We treat the provided bounding boxes as visual prompt inputs, discarding the original ground truth coordinates from the questions to create high-quality instruction data. We utilize grounding QA pairs from

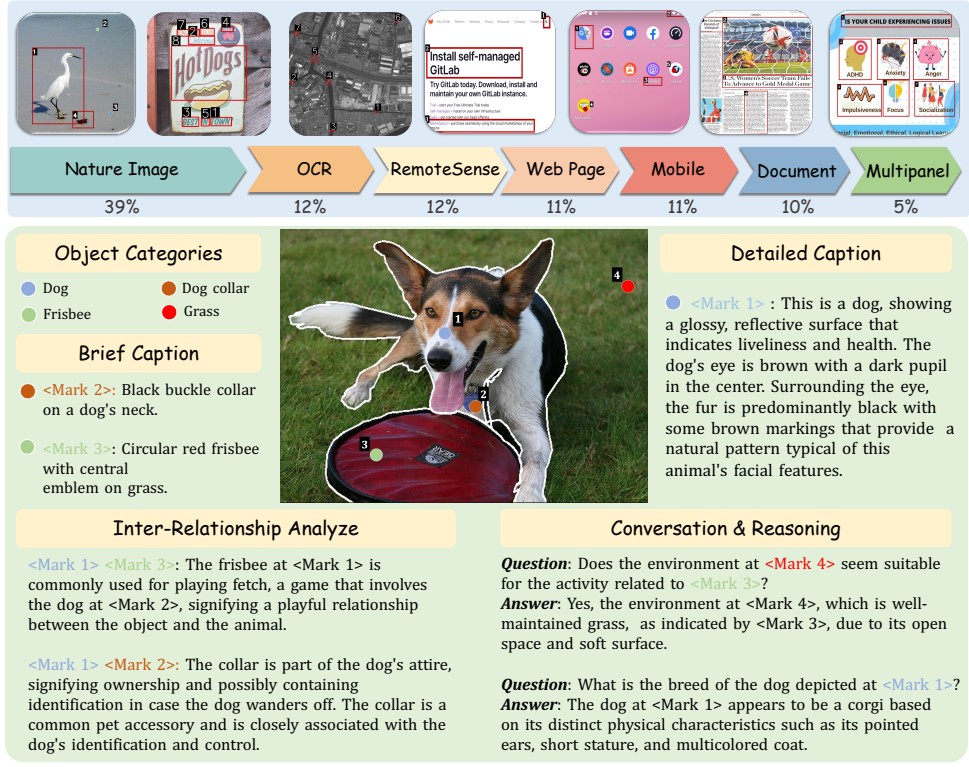

Figure 3: **An illustrative example from the MDVP-Instruct-Data.** This diagram illustrates the distribution of images sourced from various domains. It also highlights a GPT-assisted sample, emphasizing the diversity and richness of its point-based and region-based instruction-following data

the point-QA (Shim, 2003), Visual 7W (Zhu et al., 2016), and VCR (Zellers et al., 2019) datasets. *(ii) Constructing with GPT-4V Assistance*. To build a more comprehensive and diverse instruction-following dataset, we collected various multi-domain images along with their unique annotations from public datasets (Table 5). These include natural images, Optical Character Recognition (OCR) content in the wild, documents, webpage screenshots, mobile screen captures, remote sensing imagery, and multi-panel images. For each distinct image domain, we meticulously crafted prompts to facilitate GPT-4V's ability to adopt various roles in generating instructional data. Specifically, we assigned GPT-4V four distinct tasks: brief captioning, detailed captioning, inter-relationship analysis, and complex reasoning generation. Notably, the captions created for each domain exhibit unique characteristics. For instance, in mobile screen captures, objects are identified not only as relevant components—such as icons, text, search bars, and URLs—but also include information about potential actions, indicating whether a component is clickable and the expected outcome upon interaction. This diversity significantly enriches the dataset while also presenting challenges. Furthermore, as shown in Fig.3, to enhance GPT-4V's recognition of objects referenced by visual prompts, we employed Set-of-Marks (SoM) prompting (Yang et al., 2023). This method directly highlights objects in the input images, ensuring a strong association between the generated data and the referenced objects. Additionally, we propose incorporating the category information of each object within the prompts, which greatly enhances the quality of the generated data. (Details about data construction can be found in Appendix D.)

**MDVP-Bench.** To evaluate the proficiency of MLLMs in visual prompting tasks and their versatility across various domains, we initially curated a subset of our MDVP-Instruct-Data. This subset underwent a thorough manual content filtering and refinement process, resulting in the creation of MDVP-Bench. MDVP-Bench

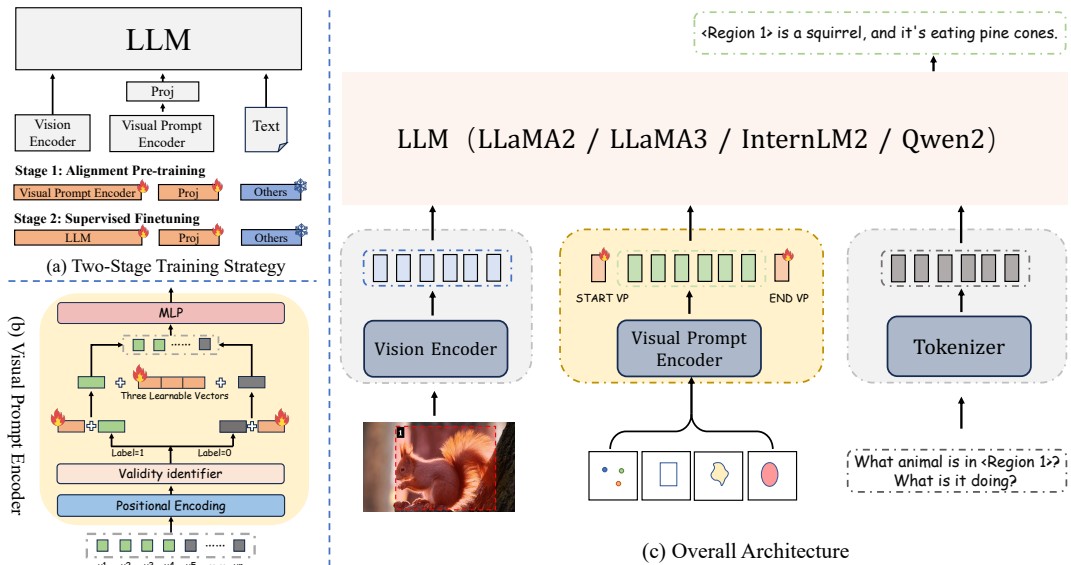

Figure 4: **(a)** The proposed training strategy for our VP-MLLM. **(b)** The detail of visual prompt encoder. **(c)** The overall architecture of the VP-MLLM. The Vision Encoder, Tokenizer, and LLM are derived from pre-trained MLLMs such as LLaVA (2024), SPHINX (2024), Qwen-VL (2023), and others. By incorporating our proposed visual prompt encoder, we can feed the image tokens, visual prompt tokens, and text tokens together into the LLM for training, equipping the MLLM with visual prompting understanding capabilities.

serves as a challenging benchmark that encompasses a wide range of tasks, including concise descriptions, elaborate narratives, analyses of interconnections between regions, and complex reasoning.

For open-ended evaluations, existing methods typically follow LLaVA's approach (Liu et al., 2024b), using GPT-4 with textual descriptions to depict image content. However, these descriptions often depend on image annotations, which can lead to situations where GPT-4 fails to recognize unannotated objects or backgrounds. Additionally, inherent domain gaps between textual descriptions and images may result in misunderstandings of the image content. To ensure a more robust evaluation, we employ GPT-4V. We directly annotate images using SoM prompting (Yang et al., 2023), submitting both the images and textual questions to GPT-4V for scoring. The scoring follows the LLaVA-bench guidelines, with scores ranging from 1 to 10, where higher scores indicate better model performance.

## 3 VISUAL PROMPTING MLLM

In this section, we detail the process of integrating visual prompt understanding into pre-trained MLLMs and transforming them into Visual Prompting MLLMs (VP-MLLMs). We also introduce a training strategy designed to enhance alignment and fine-tuning for VP-MLLMs.

### 3.1 ARCHITECTURE

**Overall Architecture of VP-MLLM.** For most existing MLLMs, the overall architecture consists of three components: a vision encoder, a tokenizer (text encoder), and a Large Language Model (LLM). Each modality is processed by its corresponding encoder, and the resulting tokens are concatenated and fed into the LLM for learning. Similarly, to achieve visual prompting understanding, we incorporate a visual prompt encoder to embed the input visual prompts, as shown in Fig. 4(c). This integration allows us to combine the image, visual prompt, and language representations and forward them collectively to the LLM.

**Visual Prompt Encoder.** As shown in Fig. 4(b), we introduce a simple yet effective visual prompt encoder that focuses on two types of visual prompts: points and bounding boxes. Initially, the encoder utilizes positional encoding(Tancik et al., 2020) for the coordinates of both points (center) and boxes (top-left and bottom-right corners). It then adds three distinct learnable embeddings for each corner and processes them through a linear layer to obtain unified output embeddings. Additionally, we accommodate a dynamic number of visual prompts as input. We first set a fixed number of visual prompt tokens (e.g., 16). Based on the validity of the actual input tokens, we provide both valid and invalid tokens with a set of learnable vectors to help the model discern their effective features. Finally, we employ a linear layer to map the embeddings of different prompt types to the same dimension, thereby unifying the various visual prompt inputs.

## 3.2 TRAINING STRATEGY FOR VP-MLLM

**Stage 1: Image-Visual Prompt-Text Alignment Pre-training.** We initially freeze both the pre-trained vision encoder and the LLM, then focus on training the features of visual prompts to align with those of the input image and text. Following the approach in LLaVA (Liu et al., 2024b), we implement an MLP to transform the visual prompt tokens into the latent space of the LLM. We use open-source detection and segmentation datasets to create our stage 1 training data. These datasets include a wide range of objects and label types, such as elements of the natural world (e.g., people, animals, objects), remote sensing (e.g., buildings, roads, vehicles, water bodies), document components (e.g., titles, paragraphs, images, tables), OCR data (e.g., text recognition), and screenshots (e.g., icons, text, search bars). For point visual prompts, we randomly sample pixels from semantic segmentation images, where each point corresponds to a pixel-level label annotation. For box visual prompts, we directly use the ground truth bounding boxes from detection datasets as inputs, enabling the model to recognize their corresponding labels. With this rich and diverse data for pre-training, the model is well-equipped for visual prompting and object categorization. The datasets used in stage 1 are shown in Appendix (Table 5).

**Stage 2: Multi-Task Instruction Finetuning.** At this stage, we load the weights trained from stage 1 and keep the vision encoder and visual prompt encoder weights frozen. We then fine-tune the visual prompt projector and the LLM. This stage focus on enhancing model's ability to accurately interpret user instructions and handle diverse visual prompting understanding tasks, such as detailed captioning, inter-relationship analysis, and complex reasoning, while maintain the original robust vision-language global understanding capability. Table 6 outlines all the data utilized during stage 2 fine-tuning, which includes our proposed MDVP-Instruct-Data, Visual Genome (VG)(2017), Visual Commonsense Reasoning (VCR)(2019), Visual7w (2016), Osprey-724k (2024a), and multiple open-source image-level instruction datasets.

**Simulation Training for Free-Form Visual Prompt Inputs.** To support free-form visual prompts, we introduce a noise-based augmentation during the alignment stage to simulate the required input area. For box prompts, Gaussian noise proportional to the box size is applied, resulting in bounding boxes that may exceed or partially cover the target, thereby approximating a free-form visual prompt's enclosing rectangle. For point prompts, we sample multiple pixels within the target's mask, guiding them toward the same object to enable precise point-based referencing. At inference, free-form inputs are pre-processed into bounding boxes, enabling flexible, user-drawn prompts.

## 4 EXPERIMENTS

### 4.1 IMPLEMENTATION DETAILS

For evaluation, we primarily use two popular MLLMs: SPHINX-X (Gao et al., 2024) and LLaVA-Next-LLaMA3 (Liu et al., 2024a), along with three model sizes: 7B, 8B, and 13B. We transform these models into VP-SPHINX and VP-LLaVA based on our proposed framework. We employ AdamW (Loshchilov & Hutter, 2017) as our optimizer and leverage flash attention (Ford et al., 2009) to enhance computational efficiency. During the stage 1 training phase, we set the starting learning rate to $4e-5$. In stage 2, the initial

Table 1: Results of referring classification on LVIS and PACO, and COCO text. Calculation of Semantic Similarity and Semantic IOU was performed using box visual prompts. We randomly perturb the center positions and scale the dimensions of the box visual prompts to simulate free-form inputs.

| Method | LVIS(2019) | | | | | PACO(2023) | | COCO Text(2016b) |
|---|---|---|---|---|---|---|---|---|
| | Semantic Similarity | Semantic IOU | Accuracy | | | Semantic Similarity | Semantic IOU | Accuracy |
| | | | Point | Box | Free-Form | | | Box |
| LLaVA-7B(2023b) | 48.95 | 19.81 | 50.1 | 50.3 | - | 42.20 | 14.56 | - |
| Shikra-7B(2023b) | 49.65 | 19.82 | 57.8 | 67.7 | - | 43.64 | 11.42 | - |
| GPT4RoI-7B(2023a) | 51.32 | 11.99 | - | 61.8 | - | 48.04 | 12.08 | - |
| ChatSpot-7B(2023) | - | - | - | 64.5 | - | - | - | 31.8 |
| Osprey-7B(2024a) | 65.24 | 38.19 | - | - | - | 73.06 | **52.72** | - |
| Ferret-13B(2023) | 64.96 | 37.82 | 68.4 | 80.5 | 71.0 | - | - | - |
| Ferret-v2-13B(2024) | - | - | 75.1 | 87.7 | 76.4 | - | - | - |
| VP-SPHINX-7B | 86.02 | 61.24 | 85.44 | 88.50 | 86.19 | 74.15 | 49.88 | 43.67 |
| VP-SPHINX-13B | **87.06** | **62.90** | **86.46** | **89.82** | **88.96** | **76.82** | 51.32 | **45.44** |
| VP-LLaVA-8B | 86.67 | 61.52 | 85.85 | 89.44 | 88.09 | 75.67 | 50.04 | 44.82 |

learning rate was adjusted to $1e - 5$. The input images were processed using each model's unique dynamic resolution mechanism, and the maximum sequence length for the Large Language Model (LLM) was set to 3072. In all evaluation experiments, we will not continue to fine-tune on a specific dataset but will instead adopt a zero-shot testing approach.

## 4.2 REFERRING CLASSIFICATION

**Object Classification.** This task is defined as follows: the question targets a specific area within the image, requiring the model to identify the object in that designated region. Following (Yuan et al., 2024a), we employ two semantic relevance indicators—Semantic Similarity (SS) and Semantic Intersection over Union (S-IOU)(Rezatofighi et al., 2019)—to assess the model's classification performance on the validation sets of the object-level LVIS(Gupta et al., 2019) and part-level PACO (Ramanathan et al., 2023) datasets. As shown in Table 1, our VP-SPHINX and VP-LLaVA significantly outperform state-of-the-art methods in terms of SS and S-IOU by a notable margin.

We also conducted tests on traditional closed-set object classification tasks. To ensure the model outputs names of categories within the closed set, we followed the approach described in (You et al., 2023) and adapted our evaluation to a binary-choice format. We posed questions such as, *"Please identify the labels of each marked region in the image. Is the region a (Class A) or a (Class B)?"*. The accuracy results presented in Table 1 indicate that our VP-MLLMs demonstrate a strong capability for accurately locating the Region of Interest (RoI) and identifying the corresponding object categories within that region.

**Regional Optical Character Recognition.** Optical character recognition (OCR) focuses on identifying text in images and is a fundamental aspect of visual entity recognition. Following the approach in (Zhao et al., 2023), we utilize the COCO-Text dataset(Veit et al., 2016a) to assess the regional text recognition capabilities of our VP-MLLMs. Using the ground-truth bounding boxes provided in the dataset annotations, we prompt the model with requests such as, "Please provide the OCR results for the marked region in the image." The VP-MLLMs then analyze and respond with the textual content present in the specified regions.

Given that most visual prompt-based models do not support OCR, we compare our results with ChatSpot under identical zero-shot settings. As shown in the right column of Table 1, our models outperform ChatSpot by more than 10%, highlighting its promising capability in regional text recognition. Moreover, as shown in Fig. 5, our VP-MLLMs not only recognize text in the specified regions but also exhibit enhanced understanding and descriptive abilities. For instance, they can describe characteristics such as font type, color, and background. This underscores our models' exceptional pixel-level understanding capabilities.

Table 2: Region-level captioning performance on the validation set of RefCOCOg and performance comparison on general MLLM benchmarks including VQA and OCR.

| Method | RefCOCOg(2016) | | | VQA$^{v2}$(2017) | MME$^P$(2024) | POPE(2024c) | SEED(2023) | TextVQA(2019) | DocVQA(2021) |
|---|---|---|---|---|---|---|---|---|---|
| | GPT-4V | METEOR | CIDEr | General VQA | | | | OCR | |
| Qwen-VL-7B(2024a) | - | - | - | 78.8 | - | - | 56.30 | 63.8 | 65.1 |
| LLaVA-Next-8B(2024a) | - | - | - | - | 1603.7 | - | - | 64.6 | 59.7 |
| SPHINX-X-13B(2024) | 40.16 | - | - | 80.2 | 1457.7 | 89.1 | 74.8 | 65.7 | 61.2 |
| Shikra-7B(2023b) | 40.97 | - | - | - | - | 58.8 | - | - | - |
| Shikra-13B(2023b) | 43.46 | - | - | 77.4 | - | 59.2 | - | - | - |
| Osprey-7B(2024a) | 77.54 | 16.6 | 108.3 | - | - | - | - | - | - |
| Ferret-v2-7B(2024) | - | - | - | 81.5 | 1510.3 | 87.8 | 58.7 | 61.7 | - |
| Ferret-v2-13B(2024) | - | - | - | **81.8** | 1521.4 | 88.1 | 61.7 | 62.2 | - |
| VP-SPHINX-7B | 84.37 | 21.0 | 138.7 | 76.8 | 1377.0 | 88.6 | 71.4 | 59.8 | 58.1 |
| VP-SPHINX-13B | **88.19** | **23.9** | **162.5** | 78.4 | 1412.2 | **88.9** | **73.1** | **63.2** | **60.1** |
| VP-LLaVA-8B | 86.67 | 22.4 | 153.6 | 81.7 | **1554.9** | 88.5 | 58.8 | 62.9 | 58.8 |

## 4.3 REFERRING REGION-LEVEL CAPTIONING

**Brief Region Description.** We provide quantitative comparisons for the region-level captioning task with using both mask- and box-based approaches. Specifically, we employ a box visual prompt and a text prompt, such as *"Please provide a brief description of each marked region in the image,"* to prompt our VP-MLLM to concisely describe the content of the targeted region. Experiments were conducted on the validation sets of RefCOCOg. Following (Yuan et al., 2024a), we use the METEOR and CIDEr scores to evaluate the semantic similarity between the generated captions and the ground truth. As the results shown in the left part of Table 2, our VP-MLLMs demonstrate superior performance compared to Kosmos-2 and Osprey.

**Detailed Region Description.** To evaluate the detailed region description capabilities, we leverage GPT-4 to measure the quality of responses for input referring regions. Following the approach used in Osprey (Yuan et al., 2024a), we sample 80 images from the RefCOCOg validation set (Yu et al., 2016) to generate detailed region captions using box visual prompts and text prompts like, *"Please provide a detailed description of each marked region in the image."* GPT-4 is then used to assess the captions generated by the MLLMs, with evaluation scores ranging from 1 to 10 and calculate the ratio of the predicted score to that of GPT-4, expressed as a percentage. The results shown in the first column of Table 2 indicate that our VP-MLLMs achieve the best performance, with VP-SPHINX-13B reaching an accuracy of 88.19%, significantly outperforming other region-based and mask-based methods. In comparison to other models of similar size, our VP-LLaVA-8B also achieves a score of 86.67%, surpassing the current SOTA method Osprey-7B by 9.13%.

## 4.4 COMPREHENSIVE ASSESSMENT

To comprehensively evaluate the effectiveness of our VP-MLLMs, we first use LLaVA-Bench to compare with previous models and assess general image-level understanding capabilities. Since the questions in LLaVA-Bench and general MLLM benchmarks are based on the entire image, we use bounding boxes that cover nearly the full image as visual prompts. The results are shown in Table 3. Our VP-MLLMs demonstrate highly competitive performance on complex reasoning tasks and significantly outperform other visual prompting MLLMs in conversation and detailed captioning tasks. Additionally, we conducted evaluations on a series of modern MLLM benchmarks. As shown in Table 2, our VP-MLLMs exhibit leading performance across image-level benchmarks such as MME, SEED-Bench, POPE, and OCRBench compared to existing visual prompting methods. Although their overall performance is slightly lower than that of the original MLLMs, we attribute this situation partly to our use of only a subset of open-source instruction data for training, which is in line with our expectations.

Furthermore, we use Ferret-Bench and our proposed MDVP-Bench to evaluate the pixel-level visual prompting capabilities of our models. Experiments are conducted on the Referring Description and Referring Reasoning tasks within Ferret-Bench, with the results presented in Table 3. In both tasks, our models present competitive performance compared to the current SOTA model, Ferret-v2. For MDVP-Bench, we combine

Table 3: Performance on the LLaVA Bench, Ferret Bench, and our MDVP Bench.

| Method | LLaVA Bench(2023b) | | | Ferret Bench(2023) | | MDVP Bench | | | | | | | |
| | Conversation | Detail Description | Complex Reasoning | Referring Description | Referring Reasoning | Natural | | OCR | | Multi-Panel | | Screenshot | |
| | | | | | | Box | Point | Box | Point | Box | Point | Box | Point |
|---|---|---|---|---|---|---|---|---|---|---|---|---|---|
| LLaVA-7B(2023b) | 85.4 | 68.3 | 92.1 | 41.4 | 31.7 | - | - | - | - | - | - | - | - |
| SPHINX-X-13B(2024) | - | - | - | 55.6 | 70.2 | - | - | - | - | - | - | - | - |
| Kosmos-2(2023) | 71.7 | 63.4 | 74.9 | 51.8 | 33.7 | - | - | - | - | - | - | - | - |
| Osprey-7B(2024a) | - | - | - | 72.2 | 67.8 | 86.4 | - | 18.33 | - | 48.56 | - | 28.84 | - |
| Ferret-7B(2023) | 84.4 | 79.4 | 96.3 | 68.7 | 67.3 | 84.92 | 81.42 | 21.24 | 12.20 | 44.42 | 32.08 | 26.40 | 10.62 |
| Ferret-13B(2023) | 85.2 | 80.9 | **96.4** | 70.6 | 68.7 | 86.67 | 83.40 | 27.68 | 12.20 | 49.83 | 37.65 | 30.92 | 14.40 |
| Ferret-v2-13B(2024) | - | - | - | 79.6 | 79.4 | - | - | - | - | - | - | - | - |
| VP-SPHINX-7B | 81.1 | 83.1 | 79.2 | 73.1 | 68.2 | 85.57 | 88.82 | 74.28 | 81.67 | 74.21 | 75.24 | 76.72 | 50.04 |
| VP-SPHINX-13B | 84.6 | **86.4** | 83.3 | 77.4 | 71.4 | **88.82** | **92.95** | **78.49** | **85.29** | **77.30** | **79.08** | **80.07** | **54.05** |
| VP-LLaVA-8B | **90.9** | 78.9 | 94.4 | 75.2 | 68.9 | 87.02 | 89.29 | 75.67 | 81.92 | 76.02 | 75.19 | 78.92 | 51.12 |

different tasks to provide a comprehensive assessment of the model's overall capabilities. As the results show, compared to Osprey and Ferret, our VP-MLLMs significantly outperformed them across multiple domains. This improvement can be attributed to training on our MDVP-Instruct-Data, which highlights the strong fine-grained understanding and reasoning abilities of our VP-MLLMs.

Notably, we observed that box prompt scores were sometimes lower than point prompt scores. After reviewing the model's responses, we found that in tasks with fewer visual prompts per image, box-based prompts performed better, while in tasks involving more visual prompts, point-based prompts showed significantly stronger performance. This finding highlights the effectiveness of our visual prompt encoder in modeling different types of visual prompts and suggests potential areas for future research and enhancement.

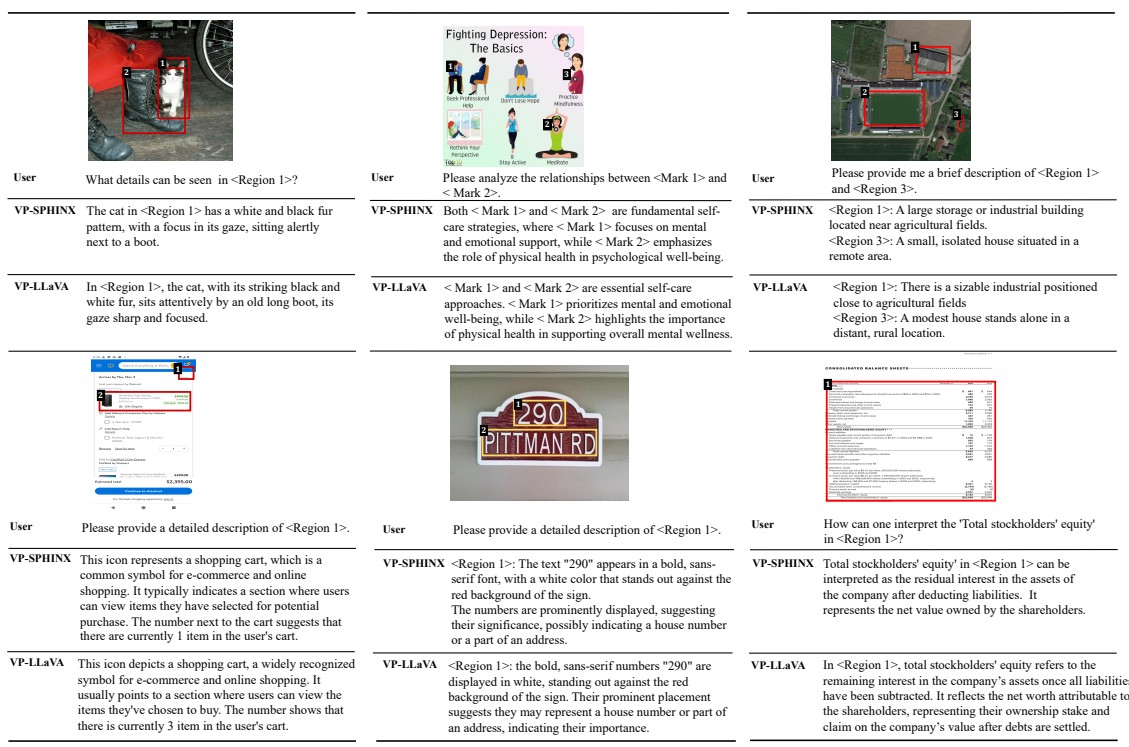

Figure 5: Qualitative examples generated by our VP-SPHINX and VP-LLaVA.

Table 4: Ablation study on different embedding format of visual prompts, training strategy, and dataset.

| Method | LVIS | | | PACO | | COCO Text | MDVP-Bench | |
| | Semantic Similarity | Semantic IOU | Accuracy | Semantic Similarity | Semantic IOU | Accuracy | Natural | |
| | | | Point | | | Box | Point | Box |
|---|---|---|---|---|---|---|---|---|
| SPHINX-13B w/ Alpha Blending | 56.89 | 33.73 | 46.84 | 40.53 | 21.14 | 21.48 | 41.19 | 64.58 |
| SPHINX-13B w/ text | 57.74 | 34.68 | 49.59 | 40.17 | 22.66 | 20.12 | 46.71 | 63.85 |
| **SPHINX-13B w/ VPE (ours)** | **68.74** | **39.29** | **69.57** | **52.73** | **27.64** | **25.81** | **67.73** | **72.54** |
| VP-SPHINX-13B w/ One-Stage | 61.13 | 33.69 | 62.94 | 46.74 | 23.96 | 21.60 | 61.24 | 65.77 |
| **VP-SPHINX-13B w/ Two-Stage (ours)** | **68.74** | **39.29** | **69.57** | **52.73** | **27.64** | **25.81** | **67.73** | **72.54** |
| **VP-SPHINX-13B w/ all MDVP data** | 68.74 | 39.29 | 69.57 | 52.73 | 27.64 | 25.81 | 67.73 | 72.54 |
| - w/ only Grounding QA data | 67.67 | 37.24 | 67.98 | 50.93 | 25.14 | 22.90 | 52.59 | 54.64 |
| - w/ only GPT4v-related data | 68.02 | 38.56 | 68.87 | 51.34 | 26.71 | 24.49 | 64.67 | 68.59 |

## 4.5 ABLATION STUDY

To evaluate the effectiveness of the key elements of our approach, we conduct the ablation experiments. Given the extensive amount of training data, our comparison was limited to the first 50k training iterations.

**The Effectiveness of Visual Prompt Encoder.** To assess the effectiveness of our visual prompt encoder (VPE), we explored two alternative methods for incorporating visual prompts into MLLMs: (1) overlaying the visual prompts on the original image via alpha blending with an alpha value of 0.5; and (2) explicitly including the coordinates of the visual prompt in the instructional text prompt. As shown in Table 4, while these alternative methods enabled the MLLM to successfully identify the regions indicated by the visual prompts, their performance was not as strong as that achieved with our visual prompt encoder. This suggests that our visual prompt encoder possesses superior capabilities for processing visual prompts.

**The Two-stage Training Strategy.** To validate the effectiveness of our proposed two-stage training strategy, we conducted an additional experiment in which we bypassed the alignment phase (stage 1) and set both the visual prompt encoder and the LLM to a trainable state for comprehensive model training. As shown in Table 4, the results demonstrate that, within the same training duration, the two-stage training approach significantly enhances visual prompting understanding, yielding improvements across all evaluated metrics.

**The Importance of GPT-4V-Constructed Data.** Since MDVP-Instruct-Data is constructed from open-source grounding QA datasets and data constructed by GPT-4V, we further validate the effectiveness of GPT-4V-related dataset. Specifically, we divide MDVP-Instruct-Data into two parts: one contains only data from the Grounding QA datasets, while the other consists solely of GPT-4V-related data. We then perform fine-tuning on each part separately. The results, as shown in Table 4, indicate that for both classification tasks and complex reasoning and dialogue tasks, the GPT-4V-constructed data provides significant improvements, underscoring its effectiveness. Moreover, when we utilize the complete dataset, the model achieve the highest performance, highlighting the overall quality of our MDVP-Instruct-Data.

## 5 CONCLUSION

In summary, we introduce a new framework, Draw-and-Understand, designed to equip existing MLLMs with robust visual prompting capabilities while preserving their original image-level perception. With our proposed visual prompt encoder, our VP-MLLMs can simultaneously support multiple types of visual prompts, including points, boxes, and free-form shapes, greatly enhancing user flexibility. Additionally, we curated the MDVP dataset, which comprises 1.2 million high-quality image-point-text and image-region-text triplets for model training, along with the comprehensive and challenging MDVP-Bench dataset for evaluation. Our superior performance in various visual prompting tasks demonstrates the efficacy and robustness of our VP-MLLMs. We believe our contributions provide a solid foundation for further exploration in the field of intelligent visual interaction systems.

## 6  ACKNOWLEDGMENTS

This project is funded in part by National Key R&D Program of China Project 2022ZD0161100, by the Centre for Perceptual and Interactive Intelligence (CPII) Ltd under the Innovation and Technology Commission (ITC)'s InnoHK, by NSFC-RGC Project N_CUHK498/24. Hongsheng Li is a PI of CPII under the InnoHK.

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

# APPENDIX

Table 5: Statistics of Training Data in Stage 1. This data also serves as the source for the GPT-4V-constructed dataset within MDVP-Instruct-Data.

| Type | Raw Data | | #Samples |
|---|---|---|---|
| Natural | COCO (Yu et al., 2016) | LVIS (Gupta et al., 2019) | 4.6M |
| | Visual Genome (Krishna et al., 2017) | OpenImages (Kuznetsova et al., 2020) | |
| | Object365 (Zhou et al., 2022) | V3Det (Wang et al., 2023) | |
| | ADE20k (Zhou et al., 2019) | Cityscapes (Cordts et al., 2016) | |
| | Pascal VOC (Vicente et al., 2014) | Flickr30k (Plummer et al., 2015) | |
| Document layout | Docbank (Li et al., 2020c) | PublayNet (Zhong et al., 2019) | 2.5M |
| | DoclayNet (Pfitzmann et al., 2022) | M6Doc (Cheng et al., 2023) | |
| OCR Spotting | ICDAR13 (Karatzas et al., 2013) | ICDAR15 (Karatzas et al., 2015) | 550K |
| | CTW1500 (Yuliang et al., 2017) | MLT2017 (Nayef et al., 2017) | |
| | MLT2019 (Nayef et al., 2019) | totaltext (Ch'ng & Chan, 2017) | |
| | CurvedSynText150k (Liu et al., 2020) | | |
| Remote Sense | DIOR (Li et al., 2020b) | DOTA (Xia et al., 2019) | 420K |
| | NWPU VHR-10 (Su et al., 2019a) | FAR1M (Sun et al., 2021) | |
| | RSOD (Zhang et al., 2023b) | HRRSD (Zhang et al., 2019) | |
| | UCAS (Xumo, 2023) | HIT-UAV (Suo et al., 2023) | |
| Android & Web | AITW (Taleby Ahvanooey et al., 2016) | SeeClick (Cheng et al., 2024) | 300K |

Table 6: Statistics of Training Data in Stage 2.

| Task | Raw Data | | #Samples |
|---|---|---|---|
| Category Identification | Subset in Stage-1 | | 1.5M |
| Brief Caption | RefCOCO (Yu et al., 2016) | RefCOCO+ (Yu et al., 2016) | 1.6M |
| | RefCOCOg (Yu et al., 2016) | Visual Genome (Krishna et al., 2017) | |
| | MDVP-Instruct-Data | | |
| Detailed Caption | MDVP-Instruct-Data | | 752K |
| Relationship Analysis | Visual Genome (Krishna et al., 2017) | RSOD (Zhang et al., 2023b) | 1.3M |
| | DIOR (Li et al., 2020b) | LEVIR (Chen & Shi, 2020) | |
| | MDVP-Instruct-Data | | |
| Multi-Target Captioning | GRIT (You et al., 2023) | Flicker30K (Plummer et al., 2015) | 1.3M |
| | OpenPsgGCG (Rasheed et al., 2023) | RRSIS-D (Yuan et al., 2024b) | |
| | GeoChat (Kuckreja et al., 2023) | | |
| Referring Q&A and Reasoning | VCR (Krishna et al., 2017) | Visual7W (Zhu et al., 2016) | 2.7M |
| | Osprey-724K (Yuan et al., 2024a) | DIOR-RSVG (Li et al., 2020b) | |
| | OPT-RSVG (Zhan et al., 2023) | MDVP-Instruct-Data | |
| General Image Q&A and Reasoning | laionGPT4v (Chen et al., 2024) | AI2D (Hiippala et al., 2020) | 4+M |
| | ChartQA (Masry et al., 2022) | DocVQA (Mathew et al., 2021) | |
| | DVQA (Kafle et al., 2018) | GeoQA (Chen et al., 2022) | |
| | LLaVA_Instruct_150k (Liu et al., 2024b) | shareGPT4v (Chen et al., 2023c) | |
| | VisualMRC (Tanaka et al., 2021) | SynthDoG (Kim et al., 2022) | |
| | MGM_Instrction (Li et al., 2024) | In-house Dataset | |

## A   IMPLEMENTATION DETAILS

### A.1   MODEL SETTINGS

To construct our VP-MLLMs, we start by inheriting the entire model structure from various pre-trained MLLMs. Specifically, for VP-SPHINX-X, we utilize the Mixture of Visual Experts (MoV)(Gao et al.,

2024) as the base image encoder and LLaMA2-7B/13B(Touvron et al., 2023) as the large language model (LLM). For VP-LLaVA-8B, we employ CLIP-ViT-L-14 ($336^2$) (Radford et al., 2021) as the base image encoder and LLaMA3-8B (Dubey et al., 2024) as the LLM. We adopt each pre-trained MLLM's respective Any Resolution strategy to process input images, effectively capturing their details. This strategy primarily involves sub-image splitting, image padding, and optimal aspect ratio selection, ultimately generating a flattened list of image tokens. Subsequently, we introduce our proposed visual prompt encoder (VPE) to process visual prompt inputs, yielding visual prompt tokens for modeling. These image tokens, visual prompt tokens, and text tokens are concatenated sequentially to form a single input for the LLM. Notably, we prepend and append a learnable token to the visual prompt tokens to serve as start and end markers.

## A.2 TRAINING DATA

Our final fine-tuning data mixture comprises a diverse range of datasets, encompassing not only a variety of image domains but also a wide array of task types. These tasks include pixel-level and region-level classification, brief captions, detailed descriptions, relationship analysis, and complex reasoning and Q&A.

**Stage 1**. Our pre-training tasks are similar to the image captioning tasks commonly used in mainstream MLLMs pre-training stage. However, they focus on learning image-level descriptions, our VP-MLLM targets pixel-level and region-level semantic classification. Specifically, we will first collect open-source datasets related to object detection, instance segmentation, and semantic segmentation. From their annotations, we can obtain ground truth pairs such as (bbox, label), (mask, label), and (pixel, label). Next, we will use bbox, mask (randomly sample a few points), and pixel (point) as visual prompts inputted into the VP-MLLM to train the model to respond with the corresponding category. During this stage, only the visual prompt encoder and the projection layer are trained. The structure of our stage-1 pre-training data is as follows:

> {"from": "human", "value": "Please identify the labels of each marked point in the image."},
> {"from": "GPT", "value": ": Label 1\n< Mark 2>: Label 2\n< Mark 3>: Label 3\n......"}

> {"from": "human", "value": "Please identify the labels of each marked region in the image. "},
> {"from": "GPT", "value": "<Region 1>: Label 1\n< Region 2>: Label 2\n< Region 3>: Label 3\n......"}

> {"from": "human", "value": "Please recognize the text of each marked region in the image. "},
> {"from": "GPT", "value": "<Region 1>: Text 1\n< Region 2>: Text 2\n< Region 3>: Text 3\n...... "}

All datasets used in stage 1 are open-source detection and segmentation datasets. A complete list of the datasets can be found in Table 5. Specifically, we collected data from five different image domains to enhance data diversity: (1) Natural Images: containing over 10k real-world semantic categories; (2) Document Images: including layout classifications such as titles, abstracts, paragraphs, and images; (3) OCR in the Wild: mainly for recognizing text in natural scenes, such as billboards and signage; (4) Remote Sensing: for identifying different regions in remote sensing images, such as playgrounds, vehicles, and pedestrians; (5) Mobile and Web Interfaces: recognizing key elements in mobile and desktop user interfaces, such as icons, text, and search bars.

**Stage 2**. In Stage 2, our primary focus is on enhancing the VP-MLLMs' ability to accurately interpret user instructions and handle diverse pixel-level understanding tasks, including detailed captioning, inter-relationship analysis, complex reasoning and so on. Table 6 provides an overview of all the data used during Stage 2 fine-tuning. Figure 6a presents an example of converting object detection data into visual prompt training data.

**Training Hyperparameters**. Both Stage 1 and Stage 2 training were conducted on 8 A100 GPUs. We show the training hyperparameters for both stage 1 vision-language-visual prompt alignment pretraining and the stage 2 instruction tuning in Table 7.

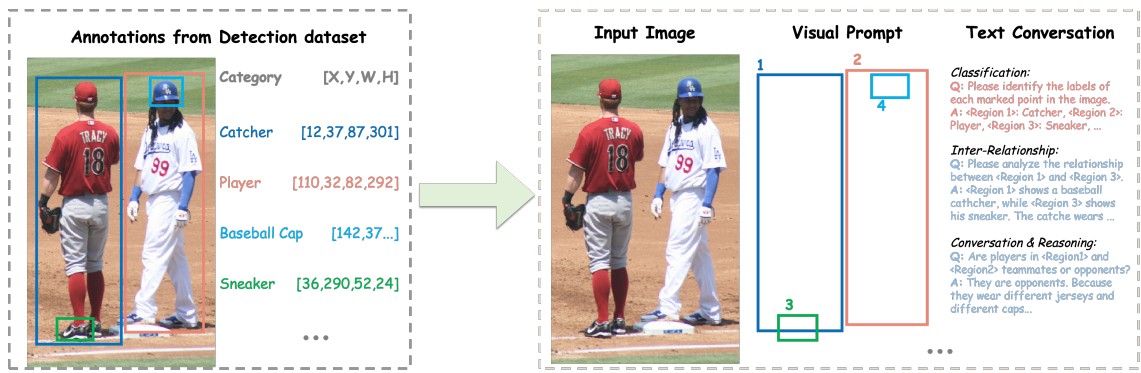

(a) An example of converting open-source object detection data into visual prompt training data. The visual prompt training data consists of three parts: the image, the visual prompt, and the text prompts (question and answer). The image is taken directly from the object detection dataset, while the visual prompt is the ground-truth bounding box with slight perturbations. The text prompts can either be selected from fixed prompt templates (e.g., the red text of Classification on the far right) or be enhanced by GPT-4V (e.g., the blue text of Inter-Relationship and Conversations & Reasoning on the far right).

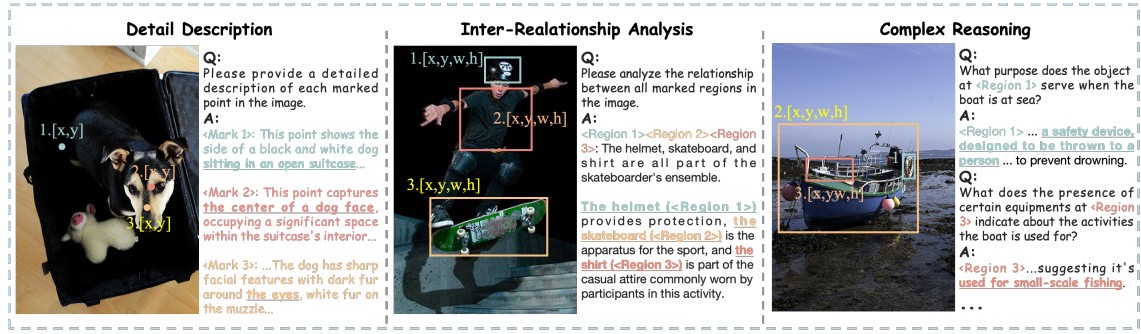

(b) Examples from the Stage 2 training data. Each visual prompt is associated with a response that focuses on describing the specific region of interest.

Figure 6: Training Data Preprocessing and Construction.

| Training Settings | Stage 1 | Stage 2 |
|---|---|---|
| Batch Size | 256 | 64 |
| Training Epochs | 1 | 1 |
| Warmup Epochs | 0.03 | 0.03 |
| Learning Rate | $4 \times 10^{-5}$ | $1 \times 10^{-5}$ |
| LR schedule | cosine decay | cosine decay |
| Gradient Clipping | 8 | 8 |
| Weight Decay | 0 | 0 |
| Optimizer | AdamW | AdamW |

Table 7: Hyperparameters of VP-MLLMs.

## B    RELATED WORKS

**Multimodal Large Language Models.** Recently, the development of Large Language Models (LLMs) has marked a significant milestone in the field of Natural Language Processing (NLP). LLMs such as the GPT series (Achiam et al., 2023), PaLM (Chowdhery et al., 2023), and LLaMA (Touvron et al., 2023) have not only achieved remarkable results in text processing but have also laid the groundwork for multimodal learning. Building on this progress, the emergence of Multimodal Large Language Models (MLLMs), including BLIP-2 (Li et al., 2023b), Flamingo (Alayrac et al., 2022), and LLaVA (Chen et al., 2024), has expanded the application scope of LLMs beyond text to other modalities. Notably, a fine-grained understanding of vision has emerged as a new focus. Works such as VisionLLM (Wang et al., 2024a) employ language-guided tokenizers to extract vision features at specific granularities, showcasing the potential in this direction.

**Visual and Multi-Modal Prompting.**   Visual and multimodal prompting in deep learning (Kirillov et al., 2023; Yuan et al., 2024a; You et al., 2023; Cai et al., 2023; Zhao et al., 2023) is an emerging area of study. Techniques utilizing visual prompts (e.g., boxes, masks) aim to enhance model performance on specific visual tasks. Key developments include SAM (Kirillov et al., 2023) and its enhanced versions, which support a broad range of prompts. Further advancements, such as SEEM (Zou et al., 2024), HIPIE (Wang et al., 2024b), and Semantic SAM (Li et al., 2023a), have improved semantic prediction. Nonetheless, for nuanced real-world applications, models require multidimensional semantic analysis, incorporating color and spatial information to fully comprehend and reason about visual scenes.

Recent advancements such as GPT4RoI (Zhang et al., 2023a), Kosmos-2 (Peng et al., 2023), Shikra (Chen et al., 2023b), Ferret (You et al., 2023), GLaMM (Rasheed et al., 2023), and ViP-LLaVA (Cai et al., 2023) have enhanced MLLMs' capabilities in region-specific image comprehension. Innovations like Colorful Prompting Tuning (CPT)(Yao et al., 2021) and RedCircle(Shtedritski et al., 2023) leverage color cues to improve model interpretative skills. Osprey (Yuan et al., 2024a) further advances this by enabling interactive, precise visual understanding using natural language and mask-based instructions. However, these methods face challenges with flexibility and complex reasoning, particularly in simultaneous multi-target referencing and their reliance on pre-defined masks, which hinders broader applications.

## C    MORE EXPERIMENTS

### C.1    REGION-LEVEL REASONING

To evaluate the reasoning capabilities of our VP-MLLMs, we utilized the Visual Commonsense Reasoning (VCR) dataset (Zellers et al., 2019), a challenging benchmark designed to assess a model's high-level cognitive and commonsense reasoning abilities within context. The VCR dataset comprises multiple-choice questions that require an understanding of the scene depicted in an image. Each question (Q) is accompanied by four possible answers (A), necessitating the model to not only identify the correct answer but also provide a rationale (R) supporting its selection. This process underscores the model's proficiency in interpreting and justifying visual elements within specific contexts, with accuracy serving as the evaluation metric. Following the evaluation approach employed in LLaVA-ViP (Liu et al., 2024a) and GPT4RoI (Zhang et al., 2023a), we fine-tuned our models using the training set of VCR. As shown in the comparison results in Tab. 8, our VP-SPHINX-13B achieves the highest performance scores of 88.92%, 90.23%, and 80.65% across three distinct evaluation methods, respectively, showcasing its proficiency in visual commonsense reasoning.

## D    MORE DETAILS OF MDVP-INSTRUCT-DATA CONSTRUCTION

In this section, we outline our methodology for reconstructing open-source grounding datasets and our application of GPT-4V to produce instruction-based data spanning a variety of domains, leading to the devel-

Table 8: Validation Accuracy on VCR dataset.

| Model | Q → A (%) | QA → R (%) | Q → AR (%) |
|---|---|---|---|
| ViLBERT (Lu et al., 2019) | 72.4 | 74.5 | 54.0 |
| Unicoder-VL (Li et al., 2020a) | 72.6 | 74.5 | 54.5 |
| VLBERT-L (Su et al., 2019b) | 75.5 | 77.9 | 58.9 |
| ERNIE-ViL-L (Yu et al., 2021) | 78.52 | 83.37 | 65.81 |
| VILLA-L (Gan et al., 2020) | 78.45 | 82.57 | 65.18 |
| GPT4RoI-7B (Zhang et al., 2023a) | 87.4 | 89.6 | 78.6 |
| ViP-LLaVA-Base-7B (Cai et al., 2023) | 87.66 | 89.80 | 78.93 |
| **VP-SPHINX-13B** | **88.92** | **90.23** | **80.65** |
| **VP-LLaVA-8B** | 88.43 | 89.96 | 79.71 |

opment of the MDVP-Instruct-Data. Fig.7 depicts an example from our dynamic multi-target captioning dataset. Sec.D.2 showcase the prompts used to generate data across these diverse domains with GPT-4V.

## D.1  EXAMPLES OF DYNAMIC MULTI-TARGET CAPTIONING DATA

Using two sample images from the Flickr30k dataset (Plummer et al., 2015) as illustrations, the Phrase Grounding task involves identifying the bounding boxes in an image that correspond to various referring phrases based on a given ground truth (GT) sentence. In a twist on this task, we invert the inputs (captions) and outputs (bounding boxes): the model is provided with bounding boxes as visual prompts and is tasked with generating natural language descriptions as responses. This process is illustrated in Fig. 7.

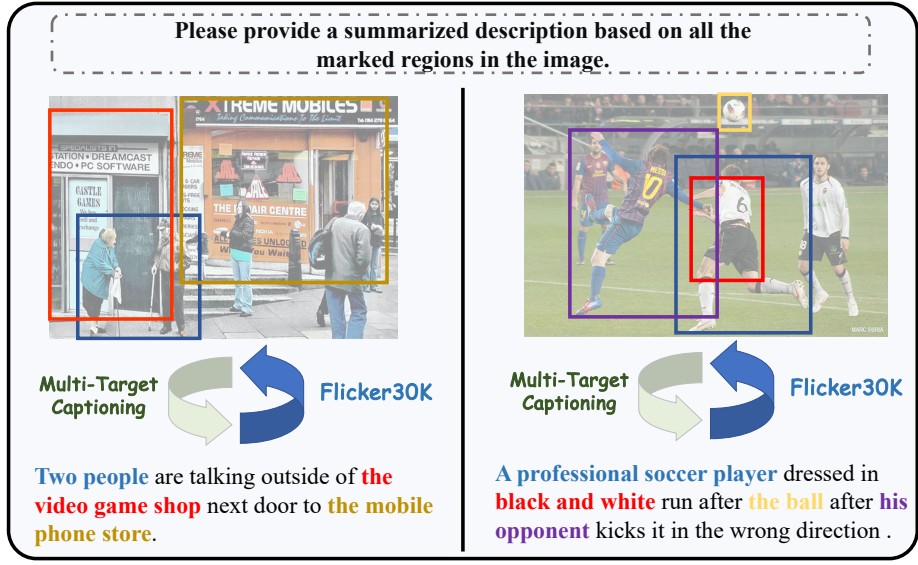

Figure 7: Examples of dynamic multi-target captioning data.

### D.2 DETAILED PROMPTS FOR GPT4V

We carefully designed prompts to facilitate data generation by GPT-4V, assigning it various roles and embedding category-specific information within these prompts. Table 9 outlines a universal prompt template applied across different domains. Meanwhile, Tables 10, 11, 12, 13, and 14 detail prompts tailored to specific domains. An illustrative example of GPT-4V's response during data generation is provided in Table 15.

## E  MORE QUALITATIVE RESULTS

We provide additional visual examples to further illustrate the pixel-level visual prompting understanding capabilities of our VP-MLLMs across a diverse range of visual prompts. Fig.8 displays a variety of visual scenarios from different domains, such as natural scenes, mobile and web interfaces, optical character recognition (OCR) in natural settings, document layouts, and multi-panel images. In these complex situations, our model showcases its proficiency in accurately responding to queries, highlighting its robust and versatile capabilities.

**image** =

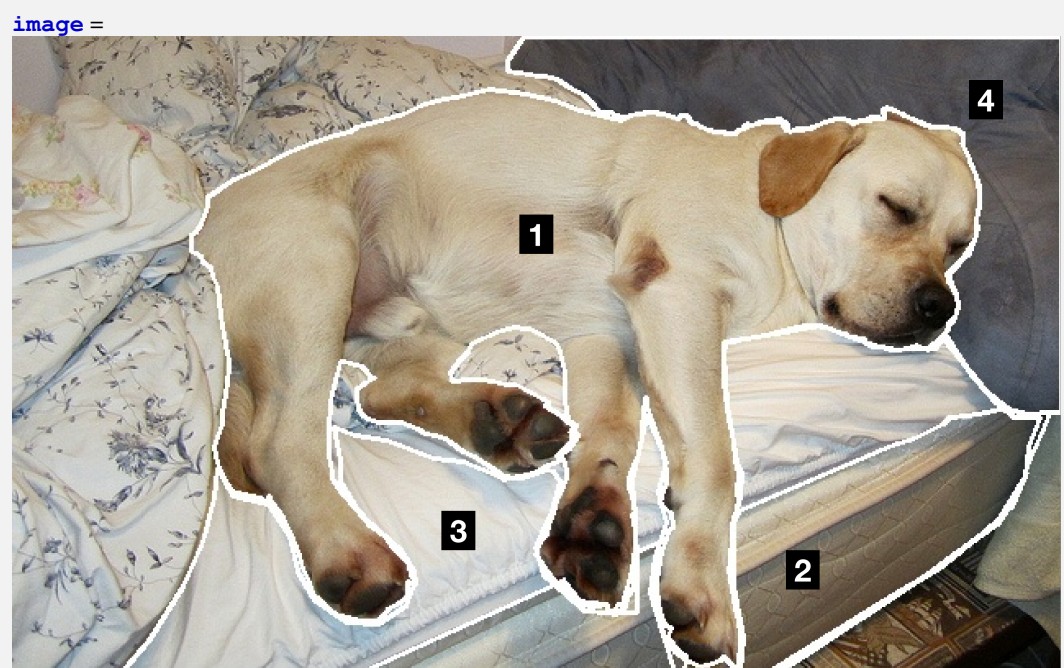

**messages** = {**"role":"system", "content":** In the image, I have marked each visual object with a green point, and each is identified by a white numeric ID against a black background.

The categories of visual objects are as follows:
**: dog**
**: bed**
**: mattress**
**: pillow**

Your analysis will revolve around four main functions:
**<Role>**

I will now supply you with specific output templates for content corresponding to the four roles. Please adhere strictly to these templates when generating content, refraining from making any additional alterations, including the insertion of extra spaces or line breaks. Adhering to this guideline is extremely crucial.
Format:
**<Format>**

Proceed with your analysis, keeping the language natural and clear.}

Table 9: The public prompt template used to feed to GPT-4V for data generation. Public prompts for different domains exhibit slight variations. For instance, in the screen-shot domain, `"In the image"` is modified to `"In the screen-shot"`. The highlighted **<Role>** and **<Format>** in red are domain-specific.

**Role**
**`<Role 1 (Short Description)>`** Provide a succinct and clear description for each marked region, ensuring each description stands independently without reference or comparison to others.
**`<Role 2 (Detailed Description)>`** For each marked region in the image, please provide as detailed a description as possible using natural language. Highlight the object's category, type, color, and additional attributes such as location, condition, and any relevant details. Envision yourself observing each region directly and convey your observations as thoroughly and promptly as possible. A minimum of 30 words is required in your description. Treat each mark as a unique and separate entity, requiring a full description exclusive to that mark only.
**`<Role 3 (Inter-Relationship Analysis)>`** Delve into and analyze the relationships between the marked regions. In your analysis, reference specific areas using identifiers like ``, ``, etc. Elucidate the links and common features among these areas, which might encompass aspects such as their spatial arrangement, inherent qualities, underlying principles, resemblances, variances, contextual ties, or notable discrepancies. Should you find certain relational aspects insignificant or lacking in noteworthy content, omit them from your discussion. In instances where a marked region stands apart without clear ties to others, highlight its distinctiveness and provide a detailed description of this unique object or area.
**`<Role 4 (Q&A and Conversations)>`** Dive deeper into the detailed content and intricacies of every marked regions, and interconnections among multiple marked regions. Employ identifiers such as ``, ``, etc., to specify each area in your inquiries and responses. Assist in formulating question-answer pairs that focus on either a single target area or multiple target areas, aiming to develop a rich dialogue dataset (comprising at least 4 Q&A pairs). Ensure that the questions you craft inquire about one or more specified ``s.
**Format**
**Role 1**
**``**: Your Short Description
. . .
**``**: Your Short Description
**Role 2**
**``**: Your Comprehensive Description
. . .
**``**: Your Comprehensive Description
**Role 3**
**``...``**: Your Detailed Analysis
. . .
**``...``**: Your Detailed Analysis
**Role 4**
{`"question"`: [Your created Question], `"Answer"`: [Your created answer]}
. . .
{`"question"`: [Your created Question], `"Answer"`: [Your created answer]}

Table 10: The **`<Role>`** and **`<Format>`** prompt template used for natural images domain.

---

**Role**

**`<Role 1 (Detailed Analysis and Description)>`** First, you need to deeply understand the page content presented by the entire screenshot, and then thoroughly examine and explain the content and purpose of the highlighted area. Provide a comprehensive description of its features, layout and functionality. Specify if the area is interactive (such as a clickable button or search bar), and describe the result or action that occurs upon interaction. If the highlighted area contains text, analyze whether it is a hyperlink, and the meaning of the text and its intended function. To ensure precise and detailed descriptions for the marked regions, please observe the following guidelines. Avoid mentioning the green rectangle or numeric ID of the marks I placed on the image, as they are not significant. Treat each mark as a unique and separate entity, requiring a full description exclusive to that mark only. A minimum of 30 words is required in your description.

**`<Role 2 (Q&A and Conversations)>`** Dive deeper into the detailed content and intricacies of every marked regions, and interconnections among multiple marked regions. Employ identifiers such as `<Region 1>`, `<Region 2>`, etc., to specify each area in your inquiries and responses. Assist in formulating question-answer pairs that focus on either a single target area or multiple target areas, aiming to develop a rich dialogue dataset (comprising at least 4 Q&A pairs). Ensure that the questions you craft inquire about one or more specified `<Region>`s.

---

Table 11: The **`<Role>`** prompt template used for screenshots domain. The **`<Format>`** can be referenced from natural images domain in Tab.10.

---

**Role**

**`<Role 1 (Detailed Region Description)>`** For each marked region in the image, provide a thorough description using natural language. First, referring to the categories of visual objects I provided you with, you need to tell me what the marked region is. Then, do your best to describe the content, characteristics, and function of the marked area. If the marked area is a text paragraph, you should first understand the content of the text, and then summarize the main idea. If the marked area is an image, you need to describe the content of the image in detail, as well as why this image is included in the document.

To ensure precise and detailed descriptions for the marked regions, please observe the following guidelines. Avoid mentioning the green rectangle or numeric ID of the marks I placed on the image, as they are not significant. Treat each mark as a unique and separate entity, requiring a full description exclusive to that mark only.

**`<Role 2 (Q&A and Conversations)>`** Dive deeper into the detailed content and intricacies of every marked regions, and interconnections among multiple marked regions. Employ identifiers such as `<Region 1>`, `<Region 2>`, etc., to specify each area in your inquiries and responses. Assist in formulating question-answer pairs that focus on either a single target area or multiple target areas, aiming to develop a rich dialogue dataset (comprising at least 4 Q&A pairs). Ensure that the questions you craft inquire about one or more specified `<Region>`s.

---

Table 12: The **`<Role>`** prompt template used for document domain. The **`<Format>`** can be referenced from natural images domain in Tab.10.

**Role**

**`<Role 1 (Detailed Region Description)>`** For each marked region in the image, I'd like you to give a detailed description, as if you were examining it in person. First refer to the OCR results of the specified area I provided to clarify the text content. Then, delve into the characteristics of the text, including the font style and its exact location within the image. I'd also appreciate insights into the background context of the region and an analysis of why the text is present—its intended purpose. Please ensure your explanation is clear and straightforward.

To ensure precise and detailed descriptions for the marked regions, please observe the following guidelines. Avoid mentioning the red polygan or numeric ID of the marks I placed on the image, as they are not significant. Treat each mark as a unique and separate entity, requiring a full description exclusive to that mark only.

**`<Role 2 (Q&A and Conversations)>`** Dive deeper into the detailed content and intricacies of every marked regions, and interconnections among multiple marked regions. Employ identifiers such as `<Region 1>`, `<Region 2>`, etc., to specify each area in your inquiries and responses. Assist in formulating question-answer pairs that focus on either a single target area or multiple target areas, aiming to develop a rich dialogue dataset (comprising at least 4 Q&A pairs). Ensure that the questions you craft inquire about one or more specified `<Region>`s.

Table 13: The **`<Role>`** prompt template used for OCR-spotting domain. The **`<Format>`** can be referenced from natural images domain in Tab.10.

**Role**

**`<Role 1 (Detailed Region Description)>`** For each marked region in the image, please provide as detailed a description as possible using natural language. Carefully observe and analyze the content and details of each panel I have marked. Tell me as best you can what the content and purpose of each marked area is, including the content of figure and the text. Envision yourself observing each region directly and convey your observations as thoroughly and promptly as possible. Treat each mark as a unique and separate entity, requiring a full description exclusive to that mark only. A minimum of 30 words is required in your description.

**`<Role 2 (Inter-Relationship Analysis)>`** Delve into and analyze the relationships between the marked regions. In your analysis, reference specific areas using identifiers like ¡Region 1¿, ¡Region 2¿, etc. Elucidate the links and common features among these areas, which might encompass aspects such as their spatial arrangement, reading order and contextual ties.

**`<Role 2 (Q&A and Conversations)>`** Dive deeper into the detailed content and intricacies of every marked regions, and interconnections among multiple marked regions. Employ identifiers such as `<Region 1>`, `<Region 2>`, etc., to specify each area in your inquiries and responses. Assist in formulating question-answer pairs that focus on either a single target area or multiple target areas, aiming to develop a rich dialogue dataset (comprising at least 4 Q&A pairs). Ensure that the questions you craft inquire about one or more specified ¡Region¿s.

Table 14: The **`<Role>`** prompt template used for multi-panels images domain. The **`<Format>`** can be referenced from natural images domain in Tab.10.

**Response**
**<Role 1 (Short Description)>**
****: Light brown dog sleeping peacefully on a bed.
****: A neatly arranged bed with light blue floral patterns.
****: Cream-colored mattress exposed at the edge of the bed.
****: Soft grey pillow partially tucked under a sleeping dog's head.
**<Role 2 (Detailed Description)>**
****: The image displays a light brown adult dog with a closed eyes and relaxed ears, lying on its side, comfortably asleep. The dog appears to be in a serene state of rest, with its head gently resting on a pillow, hinting at the trust and safety it feels in its environment.
****: This mark points to a well-kept bed covered with light blue sheets that exhibit a delicate floral design. The sheets seem smooth and unwrinkled, suggesting the bed was recently made or is well-maintained, providing a cozy resting place.
****: An exposed corner of a cream-colored mattress is visible here, revealing its thick, sturdy build. The mattress looks to be in good condition, with a soft upper layer that likely contributes to a comfortable sleeping experience.
****: Here, we see a soft grey pillow with a smooth texture. It appears to be of a thickness that provides ample support for comfort. The pillow is partially tucked under the sleeping dog's head, indicating its use for rest.
**<Role 3 (Inter-Relationship Analysis) >**
****: These marks indicate a relationship of proximity and usage. The dog **** is resting on the bed ****, which implies the bed's purpose in the household as a place for relaxation and sleep.
****: There is a direct interaction here; the dog ****)has its head on the pillow ****, showcasing the pillow's function as a support for comfort and the dog's apparent enjoyment of this provision.
****: The bed **** is made up with sheets that likely cover the mattress ****, indicating that the mattress is part of the bed setup, essential for providing a comfortable sleeping surface.
****: There is no direct relationship between the mattress **** and the pillow **** other than they both contribute to the overall comfort of the sleeping area. Each serves a different aspect of comfort and support.
**<Role 4 (Q&A and Conversations)>**
{"question": "What color is the dog at ****?", "answer": "The dog at **** is light brown."}
{"question": "Describe the design of the bed's sheets at ****.", "answer": "The sheets at **** have a light blue color with a floral design."}
{"question": "What can be said about the condition of the mattress at ****?", "answer": "The mattress at **** appears to be in good condition and sturdy."}
{"question": "How does the pillow at **** contribute to the dog's comfort?", "answer": "The pillow at **** provides support and comfort for the dog's head while it sleeps."}

Table 15: An example response from GPT-4V during data generation.

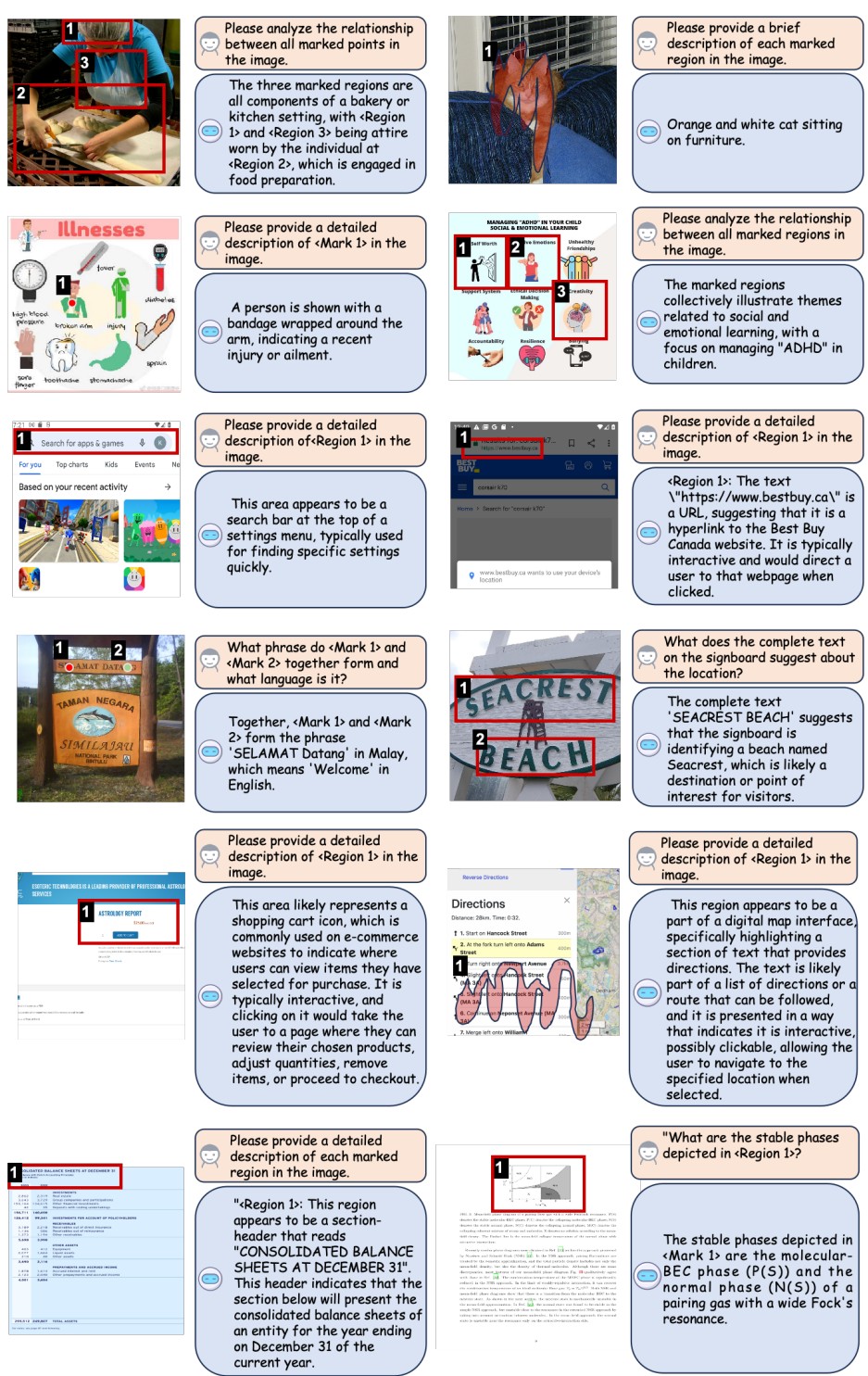

Figure 8: Additional visual examples of our VP-SPHINX-13B across multiple domains.

