# OpenReview forum: "Draw-and-Understand: Leveraging Visual Prompts to Enable MLLMs to Comprehend What You Want"
_ICLR.cc/2025/Conference — ICLR 2025 Poster_

### Official Review · Reviewer_UN1Y · 2024-11-01

**Soundness:** 3
**Presentation:** 3
**Contribution:** 3
**Rating:** 6
**Confidence:** 4

**Summary:**

The manuscript aims at improving the multi-modal LLMs’ capability of understanding user input and generating corresponding responses.
For this new task, the authors have curated a training dataset and evaluation benchmark based on open-source datasets.
For the model to be capable of handling the visual prompt input from the users, an architecture of VP-MLLM is proposed, which mainly includes an additional visual prompt encoder based on existing MLLM architecture.
The curated training data and the model architecture allows MLLM to generate responses based on point and box inputs.
The evaluation of VP-MLLM shows notable improvements on the capability of classifying and describing image contents based on specified image regions, compared to existing MLLMs.
Ablation studies demonstrate the effectiveness of the proposed visual prompt encoder and the training strategy.

**Strengths:**

- The authors have curated a training dataset as well as its corresponding evaluation benchmark for the visual prompt understanding task, which would be valuable to future research on the task.
- A new architecture for performing the visual prompt understanding task is proposed, which is shown to be effective in the experiments.
- Compared to existing approaches such as Ferret, the proposed VP-MLLM seems to be better at responding to the questions based on user inputs.

**Weaknesses:**

- It seems that the proposed approach have negative impacts on the general ability of MLLMs, as shown in Table 2.
- Lack of comparison to important baselines. It seems that the VP-LLaVA-8B introduces limited improvement over ViP-LLaVA-Base-7B (in Table 7). But the comparison is only drawn on the VCR dataset. I would be interested to see more detailed comparisons between these two approaches.

**Questions:**

- How does different design choices affect the general capabilities of the MLLM in Table 4?
- According to Sec 3, visual prompt encoder only supports points and bounding boxes. How does it accept free-form inputs as in Table 1?
- Is there any relations between the <region 1> token and the first token given to the visual prompt encoder?

---

> ### Author Response · Authors · 2024-11-22
> **Response [1/1] to Official Review by Reviewer UN1Y**
>
> > **W1: It seems that the proposed approach have negative impacts on the general ability of MLLMs, as shown in Table 2.**
>
> Thanks for pointing this out. As mentioned in line 370 of the original manuscript, this phenomenon is primarily attributed to the limited amount of image-level instruction data used for fine-tuning. To further validate this, we increased the image-level training data. The experimental results are shown in the table below.
>
> | Method | VQAv2 | MMEp | POPE | SEED | TextVQA |
> |--|:--:|:--:|:---:|:---:|:--:|
> | SPHINX-X-13B | 80.2 | **1457.7** | 89.1 | **74.8** | 65.7 |
> | VP-SPHINX-X-13B | 78.4 | 1412.2 | 88.9 | 73.1 | 63.2 |
> | VP-SPHINX-X-13B w/ more data | **80.4** | 1456.4 | **89.6** | 74.5 | **65.9**|
>
> As the results show, increasing the general image-level training data led to substantial improvements in the VP-MLLM's image-level understanding, allowing it to approach and even surpass the initial MLLM's performance on certain tasks, which demonstrates the potential of the VP-MLLM framework.
>
> > **W2: Lack of comparison to important baselines. See more detailed comparisons between ViP-LLaVA and VP-LLaVA**
>
> Thanks for pointing this out. Below, we present additional detailed comparisons between the results of ViP-LLaVA and VP-LLaVA:
>
> | Model | Classification(LVIS ) | | Visual7W | PointQA-Look Twice | | Caption (RefCOCOg) | |
> |:-:|-:|-:|--:|:---:|---|---:|---:|
> | | SS | SIoU | Acc (%) | Acc (%) | | METEOR | CIDEr |
> | ViP-LLaVA-7B | 66.73 | 40.28 | 86.60 | **71.31** | | 16.6 | 105.9 |
> | VP-LLaVA-8B (ours) | **86.67** | **61.52** | **88.30** | 66.44 | | **22.4** | **153.6** |
>
> > **Q1: How does different design choices affect the general capabilities of the MLLM in Table 4?**
>
> | Method | VQAv2 | MMEp | POPE | SEED | TextVQA |
> |---|---|---|---|---|---|
> | SPHINX-13B w/ Alpha Blending | **76.8** | **1398.6** | 87.4 | 72.2 | 61.3 |
> | SPHINX-13B w/ text | 76.5 | 1385.3 | 87.2 | **72.5** | 61.1 |
> | VP-SPHINX-13B (ours) | 76.3 | 1387.8 | **87.6** | 72.3 | **61.5** |
> | | | | | | |
> | VP-SPHINX-13B w/ One-Stage | 72.3 | 1317.5 | 81.9 | 69.2 | 57.7 |
> | VP-SPHINX-13B w/ Two-Stage (ours) | **76.3** | **1387.8** | **87.6** | **72.3** | **61.5** |
> | | | | | | |
> | VP-SPHINX-13B w/o MDVP data | 76.0 | 1378.6 | 87.2 | 72.1 | 61.5 |
> | VP-SPHINX-13B (ours) | **76.3** | **1387.8** | **87.6** | **72.3** | 61.5 |
>
> As the results shown, we can observe the following:
> - The use of alpha blending, text prompt referencing, and our visual prompt encoder yields comparable performance in terms of general capabilities.
> - The one-stage training approach significantly impacts the model’s general capabilities, likely due to the imbalance between visual prompt data and general VQA data. In one-stage training, the higher proportion of visual prompt data may lead to catastrophic forgetting of image-level knowledge.
> - The MDVP-Instruct-Data enhances image-level understanding within the VP-MLLM framework.
>
> > **Q2: According to Sec 3, visual prompt encoder only supports points and bounding boxes. How does it accept free-form inputs as in Table 1?**
>
> In Sec. 3.2 (**Simulation Training for Free-Form Visual Prompt Inputs**) of the original manuscript, we describe how free-form visual prompt inputs are handled. In summary, we simulate free-form inputs during training by adding noise to the corresponding bounding boxes. During inference, free-form inputs are preprocessed into bounding boxes for modeling. Based on experimental results and user feedback, this approach is both straightforward and effective
>
> > **Q3: Is there any relations between the <region 1> token and the first token given to the visual prompt encoder?**
>
> Yes. During the data construction, we aligned the order of <Region X> with the order of the visual prompts. <Region 1> corresponds to the first token provided to the visual prompt encoder, and <Region 2> corresponds to the second token.
>
> Thank you for your valuable feedback. We sincerely hope our response has addressed your questions and we would be glad to engage in further discussion if needed.

---

> > ### Comment · Reviewer_UN1Y · 2024-11-26
> >
> > Thanks for your response. The authors have addressed my concerns in their rebuttal, so I will maintain the current score.

---

> > > ### Author Response · Authors · 2024-11-26
> > > **Thanks for your recognition of our rebuttal!**
> > >
> > > Dear Reviewer UN1Y,
> > >
> > > Thank you for acknowledging our response and efforts! We greatly appreciate your insightful comments, which have been invaluable in improving our work.
> > >
> > > Best Regards,
> > >
> > > Draw-and-Understand Authors

---

### Official Review · Reviewer_Wihr · 2024-11-01

**Soundness:** 3
**Presentation:** 3
**Contribution:** 3
**Rating:** 6
**Confidence:** 3

**Summary:**

The paper proposes a framework for enhancing Multimodal Large Language Models (MLLMs) by integrating visual prompting capabilities. The framework allows MLLMs to interact through visual and language-based prompts, such as points, bounding boxes, and free-form shapes, enabling users to guide the model’s focus to specific image regions. Key components include a Visual Prompting MLLM (VP-MLLM) architecture, which combines an image encoder, a visual prompt encoder, and a language model. It is supported by a large multi-domain dataset, MDVP-Instruct-Data, and evaluated on a benchmark, MDVP-Bench, designed to test visual prompt comprehension. The framework shows superior performance in multimodal interaction and fine-grained visual understanding tasks.

**Strengths:**

The integration of visual prompting with MLLMs addresses limitations in user interactivity and allows nuanced image region referencing, which enhances model usability. By enabling point-based, box-based, and free-form prompts, the framework gives users greater flexibility in interacting with models, thus expanding the application scope in real-world tasks. I like the idea of interaction between human and models.

The MDVP-Instruct-Data and MDVP-Bench provide a rich variety of images and tasks, which improve the model’s versatility across different domains and promote robust model evaluation.

The VP-MLLM framework is adaptable to existing MLLMs, enabling straightforward integration of visual prompts across different models with minimal disruption to their original capabilities

**Weaknesses:**

The paper provides limited information on key implementation specifics, particularly regarding model parameter settings and data preprocessing steps. This lack of detail may impact reproducibility, making it challenging for other researchers to achieve consistent results in similar settings.

The framework's reliance on pre-trained models may limit its performance based on the initial capabilities of the base MLLMs, potentially constraining generalization across drastically different datasets or new models.

The two-stage training process and reliance on large datasets are resource-intensive, requiring considerable computational power for alignment and fine-tuning, which may not be accessible to all practitioners.

**Questions:**

How easily can the Draw-and-Understand framework be adapted to other MLLMs? Are there specific architectural features required in the MLLM for seamless integration?

The paper mentions that box prompts sometimes underperform compared to point prompts. Could you explain why this discrepancy occurs?

Could you provide information about the hardware used for your experiments? Knowing the specifications would help in understanding the computational requirements for reproducing your results.

---

> ### Author Response · Authors · 2024-11-22
> **Response [1/1] to Official Review by Reviewer Wihr**
>
> > **W1: The paper provides limited information on key implementation specifics, particularly regarding model parameter settings and data preprocessing steps.**
>
> Thank you for pointing this out. We have included additional implementation details regarding both the model parameter settings and data preprocessing steps in the revised paper (Appendix A).
> Furthermore, we will release all the code, model checkpoints, and preprocessed datasets to enhance reproducibility and make the implementation details more accessible to readers.
>
> > **W2: The framework's reliance on pre-trained models may limit its performance based on the initial capabilities of the base MLLMs, potentially constraining generalization across drastically different datasets or new models.**
>
> Thank you for pointing this out. After conducting further experiments, we believe that the performance of VP-MLLMs are not constrained by the initial base MLLMs and has the potential to surpass them. Here are the reasons:
> - We constructed a Stage 2 training dataset that excludes visual prompt (VP) data to train VP-MLLMs and evaluate whether VP data enhances general image-level understanding. The results, as shown in the following table, indicate that combining VP data with general data significantly improves general capabilities. This demonstrates that using both types of data during training holds the potential to surpass the performance of MLLMs trained solely on general image data.
> - When we increased the general image-level training data, the VP-MLLM exhibited substantial improvements in image-level understanding, approaching and even surpassing the initial MLLM’s performance on certain tasks. This suggests that the VP-MLLM framework can generalize to a broader range of datasets and has the potential to exceed the capabilities of the initial MLLM with more diverse and abundant training data.
>
> | Method | VQAv2 | MMEp | POPE | SEED | TextVQA |
> |--|:--:|:--:|:---:|:---:|:--:|
> | SPHINX-X-13B | 80.2 | **1457.7** | 89.1 | **74.8** | 65.7 |
> |-|-|-|-|-|-|
> | VP-SPHINX-X-13B w/o VP data| 77.6 | 1351.8 | 87.4 | 71.9 | 62.3 |
> | VP-SPHINX-X-13B | 78.4 | 1412.2 | 88.9 | 73.1 | 63.2 |
> | VP-SPHINX-X-13B w/ more data | **80.4** | 1456.4 | **89.6** | 74.5 | **65.9**|
>
> > **The two-stage training process and reliance on large datasets are resource-intensive, requiring considerable computational power for alignment and fine-tuning, which may not be accessible to all practitioners.**
>
> Thanks for the comments. In practice, VP-MLLMs benefit significantly from pre-trained base MLLMs, enabling rapid convergence during fine-tuning.
>
> Based on our experiments, with eight A100 GPUs, Stage 1 requires only 2 days of training, while Stage 2 takes approximately 3 days to produce a satisfactory VP-MLLM. Therefore, compared to other visual prompting MLLM frameworks, our approach is relatively more efficient and accessible for individual researchers.
>
> > **Q1: How easily can the Draw-and-Understand framework be adapted to other MLLMs? Are there specific architectural features required in the MLLM for seamless integration?**
>
> Integrating the VP-MLLM framework into other MLLMs is simple and straightforward. The steps are as follows:
> 1. Inherit the entire model structure of the initial MLLMs.
> 2. Incorporate our visual prompt encoder to handle the visual prompt inputs.
> 3. Concatenate the image tokens, visual prompt tokens, and text tokens sequentially to form a single input for the LLM.
>
> This entire framework is highly efficient and streamlined, without any additional specific architectural modifications or features.
>
> > **Q2: The paper mentions that box prompts sometimes underperform compared to point prompts. Could you explain why this discrepancy occurs?**
>
> After reviewing the responses of our VP-MLLMs, we observed that for tasks involving a greater number of visual prompts within an image, point-based prompts tend to perform better. Conversely, for tasks with fewer visual prompts, box-based prompts exhibit significantly better performance. In MDVP-Bench, a key evaluation dimension and challenge lies in handling multiple visual prompts simultaneously, contributing to this phenomenon.
>
> We think one potential reason for this is that an excessive number of box prompts can lead to overlapping regions, resulting in semantic ambiguity and making it more challenging for the model to interpret. In contrast, point prompts offer a clearer reference to individual target objects, minimizing overlap and reducing semantic ambiguity.
>
> > **Q3: Could you provide information about the hardware used for your experiments?**
>
> During training, we utilized eight 80GB A100 GPUs; for inference, we only needed one 48GB A6000 GPU. (Four A100 GPUs are enough to train a 13B VP-MLLM)
>
> Thank you for your valuable feedback. We sincerely hope our response has addressed your questions and we would be glad to engage in further discussion if needed.

---

> > ### Comment · Reviewer_Wihr · 2024-11-25
> > **Reply**
> >
> > Thank you for your comments.
> >
> > My concerns have been addressed. However, I am uncertain about the novelty of this research paper as I am not an expert in this area. Therefore, I will maintain my previous score of 6 with a confidence level of 3.

---

> > > ### Author Response · Authors · 2024-11-26
> > > **Thanks for your reply!**
> > >
> > > Dear Reviewer Wihr,
> > >
> > > Thank you for your insightful comments and the efforts! We fully appreciate and respect your perspective on the evaluation.
> > >
> > > In fact, **Draw-and-Understand** not only provides a simple yet effective unified framework that enables MLLMs to become VP-MLLMs while maintaining their original image-level understanding capabilities, but also makes a significant contribution with the creation of MDVP-Instruct-Data. This dataset is not only beneficial to our own VP-MLLMs but also to all other visual prompting MLLMs, and has already seen use in industry scenarios. We genuinely believe that Draw-and-Understand offers an unique framework and significant contributions to the visual prompting field.
> > >
> > >
> > > Best Regards,
> > >
> > > Draw-and-Understand Authors

---

### Official Review · Reviewer_SHy1 · 2024-11-04

**Soundness:** 3
**Presentation:** 3
**Contribution:** 3
**Rating:** 5
**Confidence:** 4

**Summary:**

The paper introduces a framework that enhances multimodal large language models (MLLMs) with robust visual prompting capabilities. It proposes a novel architecture, VP-MLLM, that integrates a vision encoder, a visual prompt encoder, and an LLM, allowing models to interpret diverse visual prompts like points, bounding boxes, and free-form shapes. Additionally, the authors present the MDVP-Instruct-Data, a large-scale, multi-domain dataset designed to support visual prompting and image-level perception. The experimental results, tested on MDVP-Bench, demonstrate that VP-MLLMs outperform existing methods in pixel-level understanding and multimodal interaction, enhancing MLLMs' capacity for detailed image analysis and spatial reasoning.

**Strengths:**

1.  The paper introduces an innovative visual prompt encoder and adapts existing MLLMs for enhanced visual prompting, enabling user-friendly interactions that incorporate spatial and region-specific cues.
2. The experimental results are comprehensive, covering a wide range of tasks and benchmarks. The performance on MDVP-Bench demonstrates the effectiveness of the proposed approach in terms of pixel-level understanding and multimodal interaction.
3. The paper is generally well-organized, with a logical flow from the problem statement to the methodology and results. The figures and tables support understanding.

**Weaknesses:**

1. The two-stage training strategy is presented as a key component, but details on the alignment stage (stage 1) are sparse. Additional clarification regarding the pre-training tasks and specific data used in this phase would enhance reproducibility.
2. While the paper presents ablation studies, there is limited insight into the specific impact of the visual prompt encoder's internal mechanisms. A more granular ablation of this component would clarify its effectiveness relative to simpler alternatives.
3. The model’s reliance on the quality of visual prompt data, particularly the GPT-4V-constructed dataset, raises concerns about its robustness across datasets with varying annotation quality. An analysis of performance variation based on prompt quality would be beneficial.
4. Although the framework is promising, the potential computational cost associated with handling multiple visual prompts and spatial references is not discussed. Assessing the model's performance in terms of processing time and scalability would address practical implementation concerns.

**Questions:**

1. Could the authors provide more information on the data used for alignment pre-training in stage 1? Specific details on the dataset types and the nature of the pre-training tasks would enhance reproducibility.
2. How does VP-MLLM's performance vary with lower-quality or inconsistent visual prompt data? A discussion on the impact of prompt quality on model accuracy would clarify the robustness of the proposed framework.

---

> ### Author Response · Authors · 2024-11-22
> **Response [1/2] to Official Review by Reviewer SHy1**
>
> > **Q1 and W1:  Provide more information on the data used in stage 1. Specific details on the dataset types and the nature of the pre-training tasks would enhance reproducibility.**
>
> Thanks for your comments. Here are details about the data and pre-training tasks in stage 1.
>
> **Pre-training tasks:**
>
> In stage 1, the objective is to effectively align the visual prompt features with the image and text features. Therefore, similar to the image captioning tasks commonly used in mainstream MLLMs pre-training stage, we introduce pixel-level and region-level semantic classification tasks, with only the visual prompt encoder and the projection layer are trainable.
> Specifically, we first collected open-source datasets related to object detection, instance segmentation, and semantic segmentation. From their annotations, we can obtain ground truth pairs such as (bbox, label), (mask, label), and (pixel, label). Next, we used bbox, mask (randomly sample a few points), and pixel (point) as visual prompts inputted into the VP-MLLM to train the model to respond with the corresponding category. Here are some examples of conversations used during stage 1:
>
> [{"input": "Please identify the labels of each marked `point` in the image."},
>  {"output": "`<Mark 1>`:Label 1\n`<Mark 2>`:Label 2\n..."}]
>
> [{"input": "Please identify the labels of each marked `region` in the image."},
>  {"output": "`<Region 1>`:Label 1\n`<Region 2>`:Label 2\n..."}]
>
> [{"input": "Please recognize the `text` of each marked `region` in the image."},
>  {"output": "`<Region 1>`:Text 1\n`<Region 2>`:Text 2\n..."}]
>
> **Pre-training data:**
>
> All datasets used in stage 1 are open-source detection and  segmentation datasets. A complete list of the datasets can be found in Table 5 of the Appendix. Specifically, we collected data from five different image domains to enhance data diversity:
>
> (1) Natural Images: containing over 10k real-world semantic categories;
>
> (2) Document Images: including layout classifications such as titles, abstracts, paragraphs, and images;
>
> (3) OCR in the Wild: mainly for recognizing text in natural scenes, such as billboards and signage;
>
> (4) Remote Sensing: for identifying different regions in remote sensing images, such as playgrounds, vehicles, and pedestrians;
>
> (5) Mobile and Web Interfaces: recognizing key elements in mobile and desktop user interfaces, such as icons, text, and search bars.
>
> We have provided more details on the data used in stage 1 and stage 2 in the revised paper. (Appendix A.2)
>
> > **W2: There is limited insight into the specific impact of the visual prompt encoder's internal mechanisms. A more granular ablation of this component would clarify its effectiveness relative to simpler alternatives.**
>
> To further demonstrate the effectiveness of our visual prompt encoder (VPE), we decompose it into two parts:
>
> **VPE-base** represents the fundamental components, only including positional encoding[1] and an MLP output layer;
>
> **VPE** comprises trainable parameters that enhance modeling capabilities and a validity identifier used to dynamically determine the number of input visual prompts.
>
> Based on this decomposition, we conducted an ablation study. As results shown, with only the basic component, the VPE outperforms using text alone, which highlights its effectiveness. Furthermore, by incorporating a few trainable parameters and the validity identifier, the VPE shows significant performance improvement. This enhancement not only strengthens the modeling capability of VPE but also enables the VPE to dynamically determine the effective number of input visual prompts, leading to more accurate responses.
>
> | | LVIS | | PACO | | MDVP-Bench |
> |-|---:|---:|-:|-:|--:|
> | | SS | SIoU | SS | SIoU | Natural (Box) |
> | w/ text | 57.74 | 34.68 | 40.17 | 22.66 | 63.85 |
> | VPE-base | 61.25 | 36.76 | 46.58 | 24.37 | 67.33 |
> | VPE | **68.74** | **39.29** | **52.73** | **27.64** | **72.54** |
>
> **Note***: SS is Semantic Similarity, SIoU is Semantic IoU. w/ text means converting the visual prompts' coordinates into text and combined with text prompts for referring.*
>
> [1] Fourier features let networks learn high frequency functions in low dimensional domains.

---

> > ### Author Response · Authors · 2024-11-22
> > **Response [2/2] to Official Review by Reviewer SHy1**
> >
> > > **Q2 and W3: How does VP-MLLM's performance vary with lower-quality or inconsistent visual prompt data? A discussion on the impact of prompt quality on model accuracy would clarify the robustness of the proposed framework**
> >
> > To address this, we divided the visual prompt data used in Stage 2 into three parts:
> >
> > (1) Open-source Data, including Visual Genome, RefCOCO3, GRIT, Flicker30K, VCR, and Osprey-724K;
> >
> > (2) MDVP-Instruct-Data only;
> >
> > (3) MDVP-Instruct-Data with noise perturbation. It is applied in two ways: by adding noise to the coordinates of the visual prompt, and by randomly adding or removing visual prompts.
> >
> > Based on this, we conducted four experiments: training only on open-source data, training only on MDVP-Instruct-Data, training only on MDVP-Instruct-Data with noise, and training on a mix of all data except for noise-augmented data. The results of these experiments are shown below:
> >
> > | VP-SPHINX-13B | Classification Task (LVIS) | | Captioning Task (RefCOCOg) | | Reasoning Task (MDVP Bench) |
> > |----|-----:|----:|----:|-------:|---:|
> > | | SS | SIoU | METEOR | CIDEr | Box (Natural) |
> > | (1) Open-source Data | 68.31| 39.08 | **18.6** | 118.3 | 67.8 |
> > | (2) MDVP data | **68.37** | **39.05** | 18.4 | **120.3** | **70.2** |
> > | (3) MDVP data w/ noise | 62.42 | 34.97 | 15.8 | 108.9 | 63.7 |
> > | - | - | - | - | - | - |
> > | All Data | **68.74** | **39.29** | **19.5** | **124.6** | **72.5** |
> >
> > As the results shown, we observe several findings:
> > - Comparing (1) and (2), both types of data effectively enhance the VP-MLLM with pixel-level classification and captioning abilities, indicating that using different source data does not compromise the robustness of the VP-MLLM. Training with MDVP data yields better performance in reasoning tasks, demonstrating that MDVP data contains high-quality dialogue and reasoning QA, which can further improve the model's performance.
> > - Comparing (2) and (3), it is evident that lower-quality data negatively impacts the performance of VP-MLLM, consistent with the consensus on the importance of training data quality. However, even with low-quality data, the VP-MLLM maintains a certain degree of visual prompting understanding, further proving the robustness of the framework.
> > - Training with all data demonstrates that increasing the amount of training data improves the model's performance. By combining the datasets, data diversity is enhanced, resulting in additional performance gains for the model.
> >
> > > **W4: Although the framework is promising, the potential computational cost associated with handling multiple visual prompts and spatial references is not discussed.**
> >
> > Thank you for pointing this out. Evaluating the computational cost of VP-MLLM is indeed important. Here are some additional details:
> > - The 13B VP-MLLM can be trained with a minimum of four A100 80GB GPUs, while inference requires only a single A6000 48GB GPU.
> > - Based on our experiments, on eight A100 GPUs, stage 1 requires only 2 days of training, and stage 2 takes approximately 3 days. After this, a satisfactory VP-MLLM can be achieved, which is relatively more effective and accessible compared to other visual prompting MLLM.
> > - For inference, we tested 100 images (with resolutions ranging from 512 to 768) on a single A6000 GPU. The average inference speed and memory usage results are as follows:
> >
> > | Model | Inference (tokens/s) | GPU Memory (GB) |
> > |----|:----:|:-----:|
> > | SPHINX-13B | 1.88 | 38.8 |
> > | VP-SPHINX-13B | 1.85 | 38.9 |
> > | LLaVA-Next-8B | 3.56 | 26.3 |
> > | VP-LLaVA-8B | 3.51 | 26.3 |
> >
> > As the results demonstrate, VP-MLLMs introduce negligible additional computational cost compared to the initial MLLMs, highlighting the efficiency of the VP-MLLM framework.
> >
> > Thank you for your valuable feedback. We sincerely hope our response has addressed your questions and we would be glad to engage in further discussion if needed.

---

> > > ### Author Response · Authors · 2024-11-25
> > > **Looking forward to your feedback**
> > >
> > > Dear Reviewer SHy1,
> > >
> > > Thank you again for the great efforts and valuable comments. We hope you find the response satisfactory. As the discussion phase is about to close, we are eagerly looking forward to hearing from you regarding any further feedback. We will be more than happy to address any additional concerns you may have.
> > >
> > > Best,
> > > Draw-and-Understand Authors

---

> > > > ### Author Response · Authors · 2024-12-01
> > > > **Looking forward to your feedback!**
> > > >
> > > > Dear reviewer SHy1,
> > > >
> > > > We are sorry that this message may bother you again. We sincerely hope that you could take your valuable time to read our response. Since the discussion deadline is already approaching, it would be nice of you to let us know whether our answers have solved your concerns so that we can better improve our work.
> > > >
> > > >
> > > > Best regards!
> > > >
> > > > Draw-and-Understand Authors

---

### Meta-Review · Area_Chair_a6W9 · 2024-12-20

**Metareview:**

This paper presents a new multimodal large-scale language model (MLLM) architecture, VP-MLLM, which extends the functionality of visual prompting. The model is designed to interpret visual prompts such as points, bounding boxes and free shapes, and uses a large-scale dataset called MDVP-Instruct-Data. Experimental results show that VP-MLLM outperforms existing methods in pixel-level understanding and multimodal interaction.

The strengths of this paper are as follows. This paper presents an innovative visual prompt encoder that extends existing MLLM by adapting it to visual prompts, enabling user-friendly interaction that incorporates spatial and domain-specific cues. The VP-MLLM framework is adaptable to existing MLLMs, allowing for easy integration of visual prompts across different models, while minimising the impact on the original functionality. Experimental results are comprehensive, covering a wide range of tasks and benchmarks. Performance on the MDVP bench demonstrates the effectiveness of the proposed approach in terms of pixel-level understanding and multimodal interaction.

The paper itself is well written, but lacks a discussion of the limitations of the proposed method and a discussion of future directions.

The proposed method is a simple integration framework that allows MLLM to become VP-MLLM while retaining the original image-level understanding function, and it is recognised as novel and effective. Therefore, the AC judged that this paper should be accepted.

**Additional Comments On Reviewer Discussion:**

This paper ended up with two positive reviews and one negative review. The two reviewers who gave positive ratings both stated that they were satisfied with the authors' rebuttal. The reviewer who gave a negative rating did not comment on the authors' rebuttal. The authors responded to the comments, in particular regarding the details of the alignment phase, the specific impact of the internal mechanism of the visual prompt encoder, the impact of poor quality or inconsistent visual prompt data, and the computational cost associated with processing multiple visual prompts and spatial references. The AC confirmed these comments and it appeared that they had been adequately addressed. Although this reviewer gave a negative rating, the novelty of the visual prompt encoder was recognised, so the AC accepted the paper positively.

---

### Decision · Program_Chairs · 2025-01-22

Accept (Poster)